# GuidedBridge: Training-freely Improving Bridge Models with Prior Guidance

Zehua Chen [* 1]   Yucheng Yang [* 1]   Binjie Yuan [1]   Kaiwen Zheng [1]   Jun S. Liu [1]   Jun Zhu [1]

## Abstract

Guidance methods, such as classifier-free guidance (CFG) and auto-guidance (AG), have advanced noise-to-data generation in diffusion models. Recently, bridge models have introduced a data-to-data generative process that can exploit an instructive clean prior. In this work, inspired by previous methods creating quality difference between denoising results as guidance, we propose a training-free bridge guidance method, termed Prior Guidance (PG). Specifically, we introduce a weak prior, which is unseen during bridge pre-training, hindering prior exploitation and thereby degrading denoising result. Then, we contrast it with the seen prior to highlight and enhance prior exploitation via a scaling factor. Moreover, we analyze the underlying mechanism of prior exploitation in the bridge process and design frequency-modulated prior guidance (FMPG), which tailors the guidance scale to low- and high-frequency bands coherent with bridge generative dynamics. To address prior exploitation in image in-painting, we develop a cascaded framework, CFG-FMPG, which first generates a noisy hidden representation via CFG and then exploits it as a generative prior with FMPG, fulfilling their complementary strengths without compromising inference efficiency. Experiments demonstrate that our PG methods consistently improve pre-trained bridge models across diverse image translation tasks.

## 1. Introduction

Diffusion models (Ho et al., 2020; Song et al., 2021b) have been a widely adopted framework for generative modeling (Chen et al., 2022b; Leng et al., 2022; Chen et al., 2022c; Liu et al., 2023b; Wang et al., 2024; Mo et al., 2025; Miao

et al., 2025; Jiang et al., 2025; Dai et al., 2026; Chen et al., 2026b), where they can faithfully reconstruct the target distribution from a known prior distribution, *e.g.*, standard Gaussian, with time-dependent score functions learned at model training stage. In conditional generation tasks, a guidance method, classifier-free guidance (CFG) (Ho & Salimans, 2021), was developed, which enhances condition alignment by extrapolating two denoising results, *i.e.*, an unconditional and a conditional one, at each inference step. Recently, another guidance method, auto-guidance (AG) (Karras et al., 2024), has been developed, which improves the accuracy of score estimation by extrapolating two denoising results: a result estimated with a well-trained network and a result estimated with an under-trained one. By creating quality differences in denoising results, these guidance methods have emphasized condition alignment and score accuracy, respectively, strengthening the generation quality of diffusion models in diverse generation tasks, such as image (Rombach et al., 2022; Esser et al., 2024), audio (Liu et al., 2023b; Jiang et al., 2025) and video generation (Blattmann et al., 2023; Wang et al., 2025b).

Given the noise-to-data sampling nature of diffusion models, they often suffer from an increased sampling burden in tasks that have already been provided with strong prior information (Chen et al., 2022a; 2023; Liu et al., 2024), such as image-to-image translation (Liu et al., 2023a). To address this limitation, bridge models (Liu et al., 2023a; Chen et al., 2023; Zhou et al., 2024; Zheng et al., 2025; Models, 2024; Zhang et al., 2025b) have introduced a data-to-data generative framework that is naturally aligned with such tasks. By directly exploiting the instructive information contained in clean prior representations, bridge models alleviate the sampling burden and achieve improved performance over conditional diffusion models in applications including image-to-video generation (Wang et al., 2025b), audio super-resolution (Li et al., 2025a;b) as well as signal restoration (Zhang et al., 2025a; Bolton et al., 2026).

Although diffusion guidance methods can be applied to bridge models, *e.g.*, improving text alignment with CFG in bridge-based image-to-video generation (Wang et al., 2025b), guidance methods tailored to bridge generative dynamics remain largely unexplored. To address this gap, we propose a novel **training-free** guidance method for bridge generative frameworks, termed **Prior Guidance (PG)**. Pre-

---

[*]Equal contribution
[1]Tsinghua University, Beijing, China. Correspondence to: Zehua Chen <zhc23thuml@mail.tsinghua.edu.cn>, Jun Zhu <dcszj@mail.tsinghua.edu.cn>.

*Proceedings of the 43rd International Conference on Machine Learning*, Seoul, South Korea. PMLR 306, 2026. Copyright 2026 by the author(s).

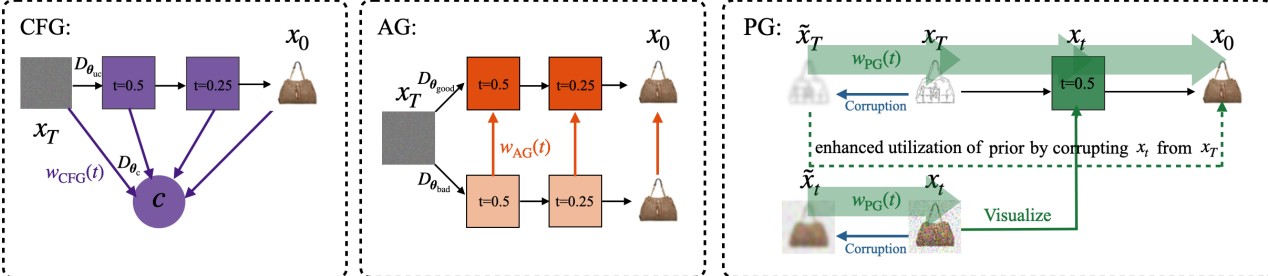

*Figure 1.* **Overview of guidance strategies.** Classifier-free guidance (CFG, left) enhances condition alignment by extrapolating an unconditional denoising result $D_{\theta_{uc}}$ and a conditional denoising result $D_{\theta_c}$. Auto-guidance (AG, middle) improves sample quality by contrasting a full-capacity denoiser $D_{\theta_{good}}$ against a less-capable denoiser $D_{\theta_{bad}}$. Our proposed prior guidance (PG, right) encourages prior exploitation by corrupting the prior representation at each bridge sampling step from $t = T$, forcing the model to strengthen its utilization of informative prior throughout the sampling trajectory. The bottom-right panel visualizes the effect of corruption on $\boldsymbol{x}_t$.

vious work shows that CFG and AG create quality differences between two denoising results to enable guidance: in CFG, unconditional denoising typically fits the data worse than conditional denoising, while in AG under-trained models produce larger errors than well-trained ones (Ho & Salimans, 2021; Karras et al., 2024; Wang et al., 2026). Motivated by the strong prior exploitation ability of bridge models, PG further amplifies this property as a guidance method to improve generation quality. Given a pre-trained bridge model that generates the target from an informative prior, we introduce an additional **weak prior**, which can be constructed via degradation operations, e.g., additional noise, blurring, or JPEG compression, in a training-free manner. As this weak prior is unseen during bridge training and provides limited target information, it increases the difficulty of prior exploitation and produces a lower-quality denoising result. We then extrapolate the two denoising results from the seen prior and the weak prior, respectively, to encourage stronger exploitation of instructive prior information at each inference step.

Furthermore, we investigate the mechanism of prior exploitation along the bridge generative process and strengthen PG through frequency modulation, termed **FMPG**. Unlike diffusion models, whose signal-to-noise ratio (SNR) increases monotonically along the noise-to-data sampling trajectory (Ho et al., 2020; Song et al., 2021b), bridge models (Liu et al., 2023a; Chen et al., 2023; Zhou et al., 2024; Zheng et al., 2025) exhibit a U-shaped SNR due to clean representations at both boundary distributions *i.e.*, data-to-data generation. Consequently, high-frequency (HF) prior information is mainly exploited at early and late sampling stages, where hidden representations are moderately corrupted by noise. In contrast, low-frequency (LF) prior information remains relatively stable even at the time step with a minimum SNR value and can therefore be exploited throughout the sampling trajectory. Based on this observation, we design FMPG, which compresses the guidance scale for HF prior

information with a U-shaped schedule while amplifying the guidance scale for LF prior information with an inverted U-shaped profile, aligning guidance with the generative dynamics of bridge models.

As discussed, given an informative prior, PG enhances prior exploitation for pre-trained bridge models by constructing a weak prior that produces a lower-quality denoising result. However, when the provided prior itself is difficult to exploit, for example, the masked region in the image inpainting task (Zheng et al., 2025), it may be difficult to construct a meaningful weak prior. To address this limitation, we propose a cascaded guidance framework, **CFG-FMPG**, which fulfills the complementary strengths of CFG and FMPG along the sampling trajectory. Specifically, at the early sampling stage, CFG reconstructs masked regions under semantic guidance, producing a noisy hidden representation that captures the coarse structure of the target. Given this representation, the guidance method is switched to FMPG to fully exploit the prior information in the following inference steps. This design assigns time-varying scaling factors to CFG and FMPG, respectively, guiding large-scale structure reconstruction with semantic guidance and high-quality refinement with prior exploitation, without sacrificing sampling efficiency.

Our contributions are summarized as follows.

- We introduce **PG**, a training-free guidance method that boosts bridge generation performance by emphasizing prior exploitation.
- We propose **FMPG**, which aligns PG with the prior exploitation mechanism of bridge generative dynamics, through frequency modulation.
- We develop **CFG-FMPG**, a cascaded framework that leverages CFG to provide coarse prior information for FMPG, thereby complementing their strengths without compromising inference speed.
- Extensive experiments demonstrate that PG methods

consistently improve pre-trained bridge models, including DDBM (Zhou et al., 2024) and DBIM (Zheng et al., 2025), across diverse image translation tasks.

## 2. Background

**Diffusion Models.** Diffusion generative framework (Ho et al., 2020; Song et al., 2021b) is composed of two mirror processes. A forward process transforms a data distribution $p_0(\boldsymbol{x}_0)$ into a known prior distribution $p_T(\boldsymbol{x}_T)$, *e.g.*, standard Gaussian distribution $\mathcal{N}(\mathbf{0}, \boldsymbol{I})$, through a stochastic differential equation (SDE) (Song et al., 2021b):

$$\mathrm{d}\boldsymbol{x}_t = \boldsymbol{f}(t)\boldsymbol{x}_t\mathrm{d}t + g(t)\mathrm{d}\boldsymbol{w}_t, \tag{1}$$

where $t \in [0, T]$, $\boldsymbol{f}$ and $g$ are predefined coefficients, $\boldsymbol{w}_t$ and $\boldsymbol{x}_t$ are the standard Wiener process and the noisy hidden representations, respectively, sharing the same data dimension with the generation target $\boldsymbol{x}_0 \in \mathbb{R}^d$. In sampling, a reverse process reconstructs the data distribution $p_0(\boldsymbol{x}_0)$ from the prior distribution $p_T(\boldsymbol{x}_T) = \mathcal{N}(\mathbf{0}, \boldsymbol{I})$ by solving a reverse SDE, which shares the same marginal distribution with $p_t(\boldsymbol{x}_t)$ defined in Equation (1):

$$\mathrm{d}\boldsymbol{x}_t = [\boldsymbol{f}(t)\boldsymbol{x}_t - g^2(t)\nabla_{\boldsymbol{x}_t}\log p_t(\boldsymbol{x}_t)]\mathrm{d}t + g(t)\mathrm{d}\bar{\boldsymbol{w}}_t. \tag{2}$$

In the model training stage, time-dependent score functions $\nabla_{\boldsymbol{x}_t}\log p_t(\boldsymbol{x}_t)$ can be learned by a denoising network $D_{\boldsymbol{\theta}}$ with:

$$\arg\min_{\boldsymbol{\theta}} \mathbb{E}_{\boldsymbol{x}_0, t, \boldsymbol{\epsilon}} \|\boldsymbol{x}_0 - D_{\boldsymbol{\theta}}(\boldsymbol{x}_t, t)\|_2^2, \tag{3}$$

where the data predictor $D_{\boldsymbol{\theta}}(\boldsymbol{x}_t, t)$ is an alternative reparameterization method of score estimation (Ho et al., 2020; Song et al., 2021b; Karras et al., 2022).

**Classifier-free guidance.** In conditional generation tasks, diffusion models can be trained with an additional network input, namely condition signal $\boldsymbol{c}$, learning the conditional denoising results $D_{\boldsymbol{\theta}}(\boldsymbol{x}_t, t, \boldsymbol{c})$. However, in practice, approximation errors caused by limited network capacity can lead to unlikely generated samples, *i.e.*, outliers, which inevitably restricts conditional generation quality (Karras et al., 2024; Wang et al., 2026). As a solution, one of the most popular methods is CFG (Ho & Salimans, 2021), a guidance algorithm that jointly models conditional and unconditional denoising results in the training stage, and then extrapolates them with a scaling parameter $w_{\mathrm{CFG}}$ at each inference step:

$$\begin{aligned} &D_{\mathrm{CFG}}(\boldsymbol{x}_t, t, \boldsymbol{c}) \\ &= D_{\boldsymbol{\theta}_{\mathrm{uc}}}(\boldsymbol{x}_t, t) + w_{\mathrm{CFG}}\left(D_{\boldsymbol{\theta}_{\mathrm{c}}}(\boldsymbol{x}_t, t, \boldsymbol{c}) - D_{\boldsymbol{\theta}_{\mathrm{uc}}}(\boldsymbol{x}_t, t)\right). \end{aligned} \tag{4}$$

Since $D_{\boldsymbol{\theta}_{\mathrm{uc}}}$ tackles a more difficult task with a small training ratio, it often fits the data distribution worse and can be viewed as a low-quality denoising result (Karras et al., 2024).

By extrapolating high-quality conditional and low-quality unconditional denoising results, CFG strengthens condition alignment and eliminates outliers, thereby improving generation results across conditional tasks (Ho & Salimans, 2021; Rombach et al., 2022; Liu et al., 2023b).

**Auto-guidance.** Recently, AG has been proposed (Karras et al., 2024), a guidance method that achieves quality improvement by extrapolating the high-quality denoising result of a well-trained model $D_{\boldsymbol{\theta}_{\mathrm{good}}}$ and the low-quality denoising result of an under-trained model $D_{\boldsymbol{\theta}_{\mathrm{bad}}}$:

$$\begin{aligned} &D_{\mathrm{AG}}(\boldsymbol{x}_t, t, \boldsymbol{c}) \\ &= D_{\boldsymbol{\theta}_{\mathrm{bad}}}(\boldsymbol{x}_t, t, \boldsymbol{c}) + w_{\mathrm{AG}}\left(D_{\boldsymbol{\theta}_{\mathrm{good}}}(\boldsymbol{x}_t, t, \boldsymbol{c}) - D_{\boldsymbol{\theta}_{\mathrm{bad}}}(\boldsymbol{x}_t, t, \boldsymbol{c})\right). \end{aligned} \tag{5}$$

Specifically, AG additionally trains an under-trained denoising network $D_{\boldsymbol{\theta}_{\mathrm{bad}}}$ on the same task, condition signal, and data distribution as the well-trained network $D_{\boldsymbol{\theta}_{\mathrm{good}}}$. These settings ensure that these two models make similar approximation errors, and $D_{\boldsymbol{\theta}_{\mathrm{bad}}}$ makes the error even stronger. Therefore, measuring and emphasizing their differences can generally indicate and lead to a direction towards better generation results (Karras et al., 2024; Wang et al., 2026).

## 3. Method

### 3.1. Motivation

As discussed, CFG and AG measure and boost quality differences arising from conditional information and model training, respectively, thereby strengthening condition alignment and score accuracy. Hence, their advantages are theoretically orthogonal to the *noise-to-data* generative framework of diffusion models (Ho et al., 2020; Song et al., 2021b) and the *data-to-data* one of bridge models (Zhou et al., 2024; Chen et al., 2023), and can be leveraged to guide bridge generation (Wang et al., 2025b). However, the key difference between diffusion and bridge models, namely *prior exploitation*, which could expand the design space for guidance strategies, has not been thoroughly explored in existing approaches. In this work, we present PG, a bridge guidance framework that creates quality differences by constructing a weak prior unseen during bridge pre-training. By further amplifying this difference and aligning it with the underlying mechanism of prior exploitation in the bridge generative process, PG explicitly emphasizes prior utilization along the sampling trajectory, leading to *training-free* improvements of bridge generation quality without sacrificing sampling efficiency (Zhou et al., 2024; Zheng et al., 2025).

### 3.2. Bridge models

**Generative framework.** Different from the *noise-to-data* diffusion generation process shown in Equation (1) and Equation (2), bridge models (Chen et al., 2023; Zhou et al.,

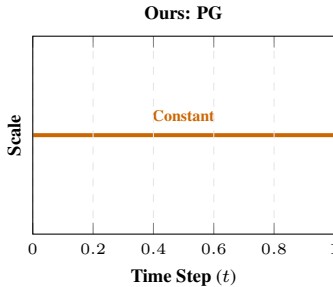 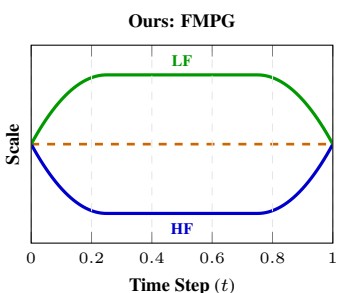 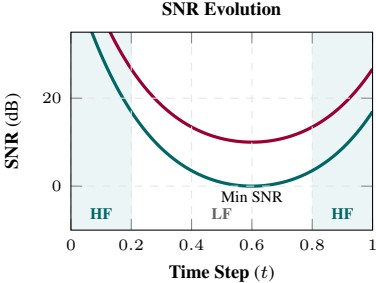

*Figure 2.* Guidance scale comparison and signal-to-noise (SNR) evolution. FMPG (Middle) adapts guidance to high-frequency (HF) and low-frequency (LF) bands, mirroring the typical U-shaped SNR (Right) observed in bridge models, whereas PG (Left) employs a constant scale and does not account for frequency dynamics.

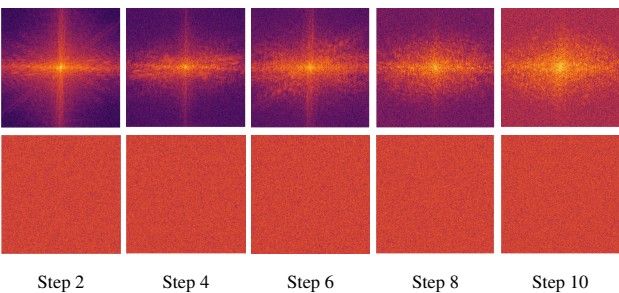

*Figure 3.* Frequency energy distribution of residuals. This plot maps the energy transfer from the input residual after extra noise addition ($\Delta \boldsymbol{x}_t$ shown in the second row of figures) to the output residual ($\Delta \boldsymbol{x}_0$ shown in the first row of figures). Brighter colors indicate higher energy.

2024; Zheng et al., 2025) learn a *data-to-data* process between a prior $p_T(\boldsymbol{x}_T) \sim p_{\text{prior}}$ and the target $p_0(\boldsymbol{x}_0) \sim p_{\text{data}}$. Specifically, conditioned on the clean prior representation $\boldsymbol{x}_T$, the forward process is modified by a drift term, ensuring that the trajectory reaches $\boldsymbol{x}_T$ at the endpoint $t = T$.

$$\mathrm{d}\boldsymbol{x}_t = [\boldsymbol{f}(\boldsymbol{x}_t, t) + g^2(t)\boldsymbol{h}(\boldsymbol{x}_t, t, \boldsymbol{x}_T)]\mathrm{d}t + g(t)\mathrm{d}\boldsymbol{w}_t, \quad (6)$$

where $\boldsymbol{h}(\boldsymbol{x}_t, t, \boldsymbol{x}_T) = \nabla_{\boldsymbol{x}_t} \log p(\boldsymbol{x}_T|\boldsymbol{x}_t)$ serves as a guiding drift derived from the pre-defined transition kernel. In generation, we reverse this process starting from $\boldsymbol{x}_T$. The time-reversed SDE is given by:

$$\mathrm{d}\boldsymbol{x}_t =[\boldsymbol{f}(\boldsymbol{x}_t, t) - g^2(t)(\boldsymbol{s}_{\boldsymbol{\theta}}(\boldsymbol{x}_t, t, \boldsymbol{x}_T) \\ - \boldsymbol{h}(\boldsymbol{x}_t, t, \boldsymbol{x}_T))]\mathrm{d}t + g(t)\mathrm{d}\bar{\boldsymbol{w}}_t, \quad (7)$$

where $\boldsymbol{s}_{\boldsymbol{\theta}}(\boldsymbol{x}_t, t, \boldsymbol{x}_T)$ is the score function approximating $\nabla_{\boldsymbol{x}_t} \log p_t(\boldsymbol{x}_t|\boldsymbol{x}_T)$. Similar to the diffusion training objective shown in Equation (3), the bridge score function can be reparameterized with data prediction $D_{\boldsymbol{\theta}}$ and trained with a denoising objective:

$$\arg \min_{\boldsymbol{\theta}} \mathbb{E}_{\boldsymbol{x}_0, \boldsymbol{x}_T, t, \boldsymbol{\epsilon}} \|\boldsymbol{x}_0 - D_{\boldsymbol{\theta}}(\boldsymbol{x}_t, t, \boldsymbol{x}_T)\|_2^2. \quad (8)$$

**Prior Exploitation.** As shown in Equation (2) and Equation (7), one of the key differences between the diffusion

and bridge generative frameworks lies in *prior exploitation*. Namely, by conditioning the marginal distribution on clean prior representation $\boldsymbol{x}_T$, bridge models disentangle the additive Gaussian noise with prior distribution, allowing strong exploitation of prior information provided by $p_T(\boldsymbol{x}_T)$ at each generation step $p_{s|t}(\boldsymbol{x}_s|\boldsymbol{x}_t, \boldsymbol{x}_T), 0 \leq s < t \leq T$. Therefore, in tasks with an informative prior, such as image-to-image translation (Liu et al., 2023a), image-to-video generation (Wang et al., 2025b), and audio super-resolution (Li et al., 2025b), bridge models can directly start sampling from the provided prior, reducing the burden of generative modeling and leading to improved results.

### 3.3. Prior Guidance

**Advantage emphasis.** Considering the advantage of bridge models over diffusion models in prior exploitation, we develop the PG method, further emphasizing this advantage to improve bridge generation quality. Given a pre-trained bridge model that has learned to generate the target $\boldsymbol{x}_0$ from the prior $\boldsymbol{x}_T$, we first construct a weak prior representation, $\mathcal{H}(\boldsymbol{x}_T)$, at the endpoint $t = T$ with a degradation operator $\mathcal{H}$. Then, at each following inference step $p_{\boldsymbol{\theta}}(\boldsymbol{x}_s|\boldsymbol{x}_t, \boldsymbol{x}_T)$, as $\boldsymbol{x}_t$ has been an updated prior for the generation target $\boldsymbol{x}_0$, we construct $\mathcal{H}(\boldsymbol{x}_t)$ as the weak prior for each hidden representation. By constructing $\mathcal{H}(\boldsymbol{x}_T)$ from the first sampling step and $\mathcal{H}(\boldsymbol{x}_t)$ along sampling trajectory, we degrade the prior information that could be exploited by the pre-trained bridge model, resulting in a degraded denoising result $D_{\boldsymbol{\theta}}(\mathcal{H}(\boldsymbol{x}_t), t, \boldsymbol{x}_T)$.

In practice, we measure the produced quality difference and extrapolate the high-quality denoising result $D_{\boldsymbol{\theta}}(\boldsymbol{x}_t, t)$ and the low-quality one $D_{\boldsymbol{\theta}}(\mathcal{H}(\boldsymbol{x}_t), t)$ with a scaling factor $w_{\text{PG}}$, to emphasize prior exploitation.

$$D_{\text{PG}}(\boldsymbol{x}_t, t, \boldsymbol{x}_T) = D_{\boldsymbol{\theta}}(\mathcal{H}(\boldsymbol{x}_t), t, \boldsymbol{x}_T) + w_{\text{PG}} \\ (D_{\boldsymbol{\theta}}(\boldsymbol{x}_t, t, \boldsymbol{x}_T) - D_{\boldsymbol{\theta}}(\mathcal{H}(\boldsymbol{x}_t), t, \boldsymbol{x}_T)), \quad (9)$$

where $t \in [T, 0)$. The condition signal $\boldsymbol{x}_T$ in the constructed low-quality term $D_{\boldsymbol{\theta}}(\mathcal{H}(\boldsymbol{x}_t), t, \boldsymbol{x}_T)$ is preserved

*Table 1.* Quantitative comparison on Edges→Handbags and DIODE datasets. We report FID (↓), IS (↑), LPIPS (↓), and MSE (↓). Our method achieves superior performance across multiple metrics.

| | | Edges→Handbags (64 × 64) | | | | DIODE-Outdoor (256 × 256) | | | |
|---|---|---|---|---|---|---|---|---|---|
| Method | NFE | FID ↓ | IS ↑ | LPIPS ↓ | MSE ↓ | FID ↓ | IS ↑ | LPIPS ↓ | MSE ↓ |
| DDIB (Su et al., 2023) | ≥ 40[†] | 186.84 | 2.04 | 0.869 | 1.05 | 242.3 | 4.22 | 0.798 | 0.794 |
| SDEdit (Meng et al., 2022) | ≥ 40 | 26.5 | 3.58 | 0.271 | 0.510 | 31.14 | 5.70 | 0.714 | 0.534 |
| Pix2Pix (Isola et al., 2017) | 1 | 74.8 | 3.24 | 0.356 | 0.209 | 82.4 | 4.22 | 0.556 | 0.133 |
| I²SB (Liu et al., 2023a) | ≥ 40 | 7.43 | 3.40 | 0.244 | 0.191 | 9.34 | 5.77 | 0.373 | 0.145 |
| DDBM (Zhou et al., 2024) | 118 | 1.83 | **3.73** | 0.142 | 0.040 | 4.43 | **6.21** | 0.244 | 0.084 |
| DDBM (Zhou et al., 2024) | 200 | 0.88 | 3.69 | 0.110 | 0.006 | 3.34 | 5.95 | 0.215 | 0.020 |
| DBIM (Zheng et al., 2025) | 20 | 1.74 | 3.63 | 0.095 | 0.005 | 4.99 | 6.10 | 0.201 | 0.017 |
| **DBIM+FMPG (Ours)** | 20 | 1.07 | 3.69 | **0.093** | **0.005** | 3.20 | 6.09 | 0.199 | **0.017** |
| DBIM (Zheng et al., 2025) | 100 | 0.91 | 3.62 | 0.100 | 0.006 | 2.58 | 6.06 | 0.198 | 0.018 |
| **DBIM+FMPG (Ours)** | 100 | **0.78** | 3.66 | 0.101 | **0.005** | **2.06** | 6.03 | **0.197** | **0.017** |

as a clean representation without degradation, which controls the same condition information as the high-quality term $D_{\boldsymbol{\theta}}(\boldsymbol{x}_t, t, \boldsymbol{x}_T)$, ensuring that the quality difference is caused by prior degradation.

**Degradation operator.** Different from AG (Karras et al., 2024) that requires an additional network to make stronger errors than the good model, PG can construct $\mathcal{H}(\boldsymbol{x}_t)$ in a training-free manner, where the degradation operator $\mathcal{H}$ can be selected from a group of methods, such as additional noise injection, blurring, and JPEG compression. In this work, considering the Gaussian transition kernel of bridge models (Liu et al., 2023a; Zhou et al., 2024; Zheng et al., 2025; Zhang et al., 2025b), we align $\mathcal{H}$ with it, namely, adding extra Gaussian noise $\boldsymbol{\epsilon} \sim \mathcal{N}(\boldsymbol{0}, \boldsymbol{I})$ to the clean prior representation $\boldsymbol{x}_T$ and each generated noisy hidden representation $\boldsymbol{x}_t$.

As shown in Figure 3, Gaussian noise in the time domain corresponds to a flat power spectral density in the frequency domain, meaning that it injects equal expected energy across all frequency bands. Consequently, adding extra Gaussian noise to the clean prior and the noisy hidden representations uniformly corrupts all spectral components of the signal, effectively weakening the ability of pretrained bridge models to exploit prior information and producing low-quality denoising result. Empirically, we find that PG effects are robust to other degradation methods, such as blurring and JPEG compression. These additional experimental results are provided in Appendix B.

### 3.4. Frequency-modulated Prior Guidance

Building upon our designed PG, we further investigate the underlying mechanism of prior exploitation in the bridge generative process and enhance PG with frequency modulation, named **FMPG**. As discussed earlier, a key difference between diffusion and bridge models lies in their exploitation of clean prior representations. Consequently, while diffusion models, *i.e.*, noise-to-data generation, exhibit a monotonically increasing SNR along the sampling trajec-

tory (Ho et al., 2020; Song et al., 2021b), bridge models, *i.e.*, data-to-data generation, demonstrate a U-shaped SNR profile (Liu et al., 2023a; Zhou et al., 2024; Chen et al., 2023), where the SNR decreases in the early sampling stage and gradually increases in late-stage sampling steps. From the perspective of prior exploitation, this U-shaped SNR trajectory implies that sampling steps close to $\boldsymbol{x}_T$ and $\boldsymbol{x}_0$ operate under relatively high SNR conditions. As a result, both the LF structural information and the HF details are more accessible for effective prior exploitation at these stages.

Figure 3 illustrates how the corruption introduced by additional Gaussian noise on $\boldsymbol{x}_t$ propagates to $\boldsymbol{x}_0$. We observe that during the intermediate time steps, where the SNR is relatively low, the additional noise on the HF components does not propagate from the prior $\boldsymbol{x}_t$ to the denoising result $\boldsymbol{x}_0$. This indicates that the model itself cannot effectively exploit the HF information at these steps, since it has already been severely corrupted by the large noise in the bridge process. In contrast, at the time steps on both sides with high SNR, the change in $\boldsymbol{x}_0$ resembles the noise injected at $\boldsymbol{x}_t$ and spans the full frequency spectrum, indicating that the model exploits the prior information well across all frequency bands at these steps.

Motivated by this underlying mechanism, we design FMPG, rescaling PG by assigning a time-varying scaling factor, $w_{\text{PG}}^{\text{LF}}$ and $w_{\text{PG}}^{\text{HF}}$, for the LF and HF bands, respectively, as shown in Figure 2. For the LF band, we have:

$$D_{\text{FMPG}}^{\text{LF}}(\boldsymbol{x}_t, t, \boldsymbol{x}_T) = I^{\text{LF}}\left[D_{\boldsymbol{\theta}}(\mathcal{H}(\boldsymbol{x}_t), t, \boldsymbol{x}_T)\right] + w_{\text{PG}}^{\text{LF}}$$
$$\left(I^{\text{LF}}\left[D_{\boldsymbol{\theta}}(\boldsymbol{x}_t, t, \boldsymbol{x}_T)\right] - I^{\text{LF}}\left[D_{\boldsymbol{\theta}}(\mathcal{H}(\boldsymbol{x}_t), t, \boldsymbol{x}_T)\right]\right),$$
$$(10)$$

where $I^{\text{LF}}$ is the LF component extracted with a low-pass filter, and $w_{\text{PG}}^{\text{LF}}$ has an inverted U-shape profile to strengthen the prior exploitation in LF band during intermediate sampling steps. For the HF band, we have

$$D_{\text{FMPG}}^{\text{HF}}(\boldsymbol{x}_t, t, \boldsymbol{x}_T) = I^{\text{HF}}\left[D_{\boldsymbol{\theta}}(\mathcal{H}(\boldsymbol{x}_t), t, \boldsymbol{x}_T)\right] + w_{\text{PG}}^{\text{HF}}$$
$$\left(I^{\text{HF}}\left[D_{\boldsymbol{\theta}}(\boldsymbol{x}_t, t, \boldsymbol{x}_T)\right] - I^{\text{HF}}\left[D_{\boldsymbol{\theta}}(\mathcal{H}(\boldsymbol{x}_t), t, \boldsymbol{x}_T)\right]\right),$$
$$(11)$$

where $I^{\text{HF}}$ is the remaining HF component, and $w_{\text{PG}}^{\text{HF}}$ has a

*Table 2.* Quantitative comparison on Edges→Handbags and DIODE datasets. We report FID ($\downarrow$), demonstrating that our guidance method outperforms baseline inference methods across both datasets given the same NFE.

| Method | Checkpoint | Edges→Handbags (NFE) | | | | DIODE (NFE) | | | |
|---|---|---|---|---|---|---|---|---|---|
| | | 10 | 20 | 40 | 100 | 10 | 20 | 40 | 100 |
| DDBM (Zhou et al., 2024) | DDBM | 137.15 | 46.74 | 7.79 | 2.40 | 151.93 | 41.03 | 15.19 | 3.34 |
| DBIM (Zheng et al., 2025) | DDBM | 2.49 | 1.74 | 1.26 | 0.91 | 7.99 | 4.99 | 3.35 | 2.58 |
| ECSI (Zhang et al., 2025b) | DDBM | 2.25 | 1.54 | - | - | 6.83 | 4.12 | - | - |
| **DBIM+FMPG (OURS)** | DDBM | **1.42** | **1.07** | **0.89** | **0.78** | **5.28** | **3.20** | **2.62** | **2.06** |

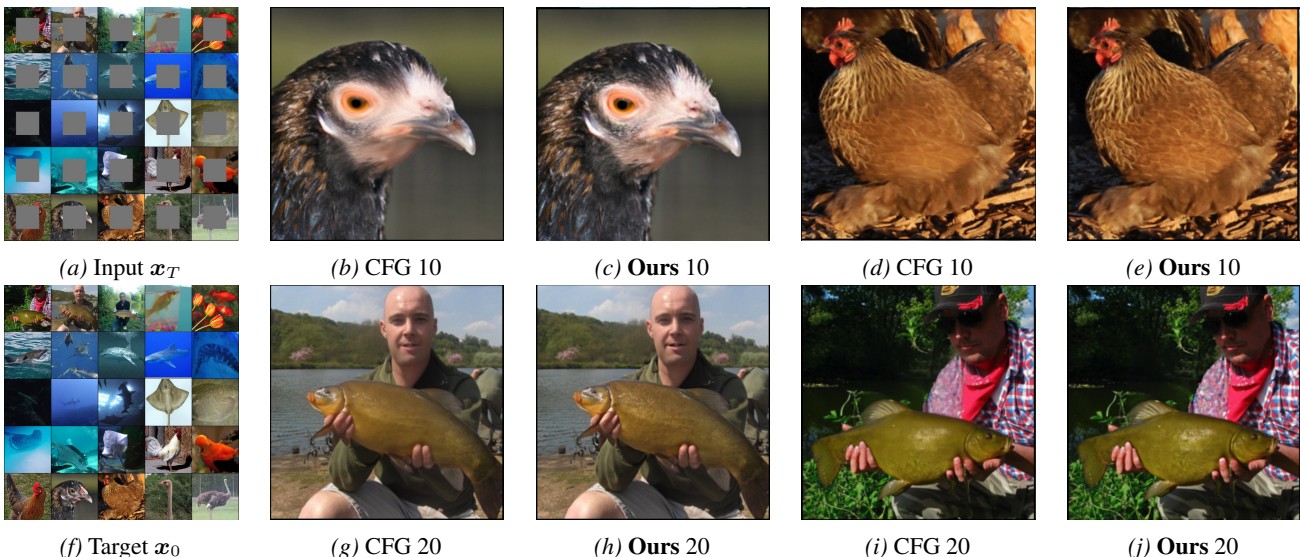

*(a)* Input $\boldsymbol{x}_T$     *(b)* CFG 10     *(c)* **Ours** 10     *(d)* CFG 10     *(e)* **Ours** 10

*(f)* Target $\boldsymbol{x}_0$     *(g)* CFG 20     *(h)* **Ours** 20     *(i)* CFG 20     *(j)* **Ours** 20

*Figure 4.* **Qualitative comparison on ImageNet restoration.** The corruption is simulated using a center **128×128 mask**. Our proposed hybrid guidance strategy recovers **semantic layout** and **high-frequency details**, simultaneously. Notably, our method outperforms the standard CFG baseline.

U-shaped profile similar to the trend of bridge SNR. Namely, considering that the prior information on the HF band has been corrupted at the time steps with a SNR value, we compress PG in the HF band during the intermediate sampling steps, which is aligned with the mechanism of prior exploitation in the bridge generative process.

Empirically, we find that a U-shaped design for $w_{\text{PG}}^{\text{HF}}$ and $w_{\text{PG}}^{\text{LF}}$ has already been able to improve the generation result compared to the constant guidance scale, and it does not require a precise alignment with the pre-defined SNR of pre-trained bridge models.

### 3.5. Integration with Classifier-free Guidance

The designs of PG methods emphasize the advantage of bridge generative process on tasks with strong prior. However, in tasks such as image in-painting, the provided information may not be a strong prior for the generation target, as shown in Figure 4. Therefore, when constructing a weak version of prior, the quality of denoising result may not be distinctively decreased, leading to marginal improvement. In this scenario, PG methods can naturally be integrated with

CFG, which provides complementary advantages. Specifically, at the early sampling stage $t \in [T, t_s)$, the prior $\boldsymbol{x}_t$ has not been able to provide an instructive prior for the generation target $\boldsymbol{x}_0$. Hence, CFG can be leveraged to guide the in-painting process with global semantic information, *e.g.*, image class $\boldsymbol{l}$, generating a noisy representation already containing large-scale features of $\boldsymbol{x}_0$ at the time step $t_s$:

$$D_{\text{CFG}}(\boldsymbol{x}_t, t, \boldsymbol{l}, \boldsymbol{x}_T) = D_{\boldsymbol{\theta}}(\boldsymbol{x}_t, t, \boldsymbol{x}_T) + w_{\text{CFG}} \\ (D_{\boldsymbol{\theta}}(\boldsymbol{x}_t, t, \boldsymbol{l}, \boldsymbol{x}_T) - D_{\boldsymbol{\theta}}(\boldsymbol{x}_t, t, \boldsymbol{x}_T)), \quad (12)$$

where $t \in [T, t_s)$. In the following sampling steps $t \in [t_s, 0)$, PG methods can stand on the CFG-guided result, emphasizing prior exploitation rather than continually strengthening the semantic alignment with CFG:

$$D_{\text{PG}}(\boldsymbol{x}_t, t, \boldsymbol{l}, \boldsymbol{x}_T) = D_{\boldsymbol{\theta}}(\mathcal{H}(\boldsymbol{x}_t), t, \boldsymbol{l}, \boldsymbol{x}_T) + w_{\text{PG}} \\ (D_{\boldsymbol{\theta}}(\boldsymbol{x}_t, t, \boldsymbol{l}, \boldsymbol{x}_T) - D_{\boldsymbol{\theta}}(\mathcal{H}(\boldsymbol{x}_t), t, \boldsymbol{l}, \boldsymbol{x}_T)). \quad (13)$$

By cascading CFG and FMPG methods along the sampling trajectory, the advantages of condition alignment and prior exploitation can be fulfilled concurrently, without compromising the inference speed (Wang et al., 2026).

# 4. Experiment

## 4.1. Experimental setting

**Datasets and Tasks.** We evaluate our method on three diverse image-to-image translation and restoration benchmarks. Edges→Handbags (Isola et al., 2017) is an edge-to-image synthesis task containing 138,567 training images. We evaluate on the entire dataset to ensure statistical significance. Then, following the experimental protocols of DDBM (Zhou et al., 2024) and DBIM (Zheng et al., 2025), we utilize the Outdoor subset of the DIODE dataset (Vasiljevic et al., 2019) for scene restoration. We perform evaluation on the complete outdoor validation split to ensure strict consistency with the baselines. ImageNet ($256 \times 256$) (Deng et al., 2009) is a large-scale class-conditional generation task. For the inpainting experiments, we apply a center mask of $128 \times 128$ to evaluate the model's ability to handle high-resolution semantic restoration.

**Backbones and Baselines.** We implement our guidance strategies on top of two state-of-the-art diffusion bridge models. DDBM (Zhou et al., 2024) is a representative bridge framework that constructs a diffusion bridge process between two arbitrary distributions. DBIM (Zheng et al., 2025) is an accelerated sampling method for bridge models, analogous to DDIM (Song et al., 2021a) in standard diffusion. For DDBM, we employ the recommended hybrid SDE-ODE sampler. For DBIM, we adopt the pure ODE mode with $\eta = 0.0$. For fair comparison, we utilize the official pre-trained checkpoints for both backbones without any modifications. We compare our methods with popular and strong baseline methods including DDIB (Su et al., 2023), SDEdit (Meng et al., 2022), PIX2PIX (Isola et al., 2017), I2SB (Liu et al., 2023a), and recently proposed ECSI (Zhang et al., 2025b).

**Corruption Methods and Guidance Scale.** When training-freely employing PG methods in the generation process, we use a two-stage tuning strategy. In the first stage, we search an appropriate corruption method, such as noise addition, blurring, JEPG compression, and pooling. In the second stage, we tune an appropriate guidance scale, which is commonly required by guidance methods, such as CFG or AG. We provide an investigation of different corruption methods in Appendix. Empirically, simple noise addition has been a robust method for image translation tasks.

**Evaluation Metrics.** We evaluate generation quality using FID (Heusel et al., 2017) and IS (Salimans et al., 2016) for distributional distance and diversity, and also reporting LPIPS (Zhang et al., 2018) and MSE to assess reconstruction fidelity. To ensure a fair comparison, we align our evaluation protocol with DBIM (Zheng et al., 2025). We also provide results across a broader range of the number of function evaluations (NFE) to comprehensively evaluate efficiency.

*Table 3.* The effect of applying PG to DDBM sampler on the Edges→Handbags dataset. We report FID ($\downarrow$) given a large number of inference steps. Noise addition is employed as the degradation method of PG.

| | NFE | |
|---|---|---|
| METHOD | 150 | 300 |
| DDBM (ZHOU ET AL., 2024) | 1.30 | 0.65 |
| **DDBM+PG** | **1.23** | **0.59** |

*Table 4.* Ablation study comparing PG and FMPG with DBIM sampler as the baseline on the DIODE dataset. We report FID ($\downarrow$) across NFEs.

| | NFE | | | |
|---|---|---|---|---|
| METHOD | 10 | 20 | 40 | 100 |
| DBIM (ZHENG ET AL., 2025) | 7.99 | 4.99 | 3.35 | 2.58 |
| ECSI (ZHANG ET AL., 2025B) | 6.83 | 4.12 | - | - |
| DBIM+PG (BLUR) | 7.33 | 3.89 | 2.64 | **2.06** |
| DBIM+PG (NOISE) | 6.25 | 3.77 | 2.96 | 2.63 |
| **DBIM+FMPG(NOISE)** | **5.28** | **3.20** | **2.62** | 2.37 |

*Table 5.* Ablation study comparing PG and FMPG with DBIM sampler as the baseline on the Edges→Handbags (**10000 images**) dataset. We report FID ($\downarrow$) across NFEs.

| | NFE | | | | |
|---|---|---|---|---|---|
| METHOD | 6 | 10 | 20 | 40 | 100 |
| DBIM (ZHENG ET AL., 2025) | 4.86 | 4.23 | 3.69 | 3.38 | 3.16 |
| ECSI (ZHANG ET AL., 2025B) | **4.10** | 3.90 | 3.59 | - | - |
| DBIM+PG(BLUR) | 4.72 | 4.01 | 3.50 | 3.26 | 3.15 |
| DBIM+PG(NOISE) | 4.86 | 3.74 | 3.44 | 3.34 | 3.20 |
| DBIM+FMPG(BLUR) | 4.27 | 3.75 | 3.36 | 3.20 | 3.14 |
| **DBIM+FMPG(NOISE)** | 4.54 | **3.51** | **3.23** | **3.13** | **3.08** |

In our experiments, we keep the NFE exactly the same as the original baseline, namely using fewer sampling steps in the guided sampling process than the generation process without guidance.

## 4.2. Main Results

**Generation Quality.** Given the same NFE, namely with fewer sampling steps as a guidance method, as shown in Table 1, DBIM+FMPG substantially improves DBIM (Zheng et al., 2025) and outperforms the baseline methods, achieving superior image translation quality. As shown in Table 2, DBIM+FMPG achieves higher quality than simply using fast bridge samplers, DBIM and ECSI (Zhang et al., 2025b), across different NFEs.

Furthermore, we validate the ability of PG to enhance the generation quality on the DDBM sampler, which prioritizes fidelity over speed. As shown in Table 3, PG consistently improves the generation quality of DDBM sampler when using a large number of inference steps.

**Sampling Efficiency.** As shown in Table 4 and Table 5, we further evaluate the efficiency of our proposed PG methods, where we systematically compare both PG and FMPG with

*Table 6.* Ablation study on ImageNet dataset comparing different guidance strategies: (1) baseline DBIM sampler without guidance, (2) full CFG guidance, (3) FMPG-first hybrid, and (4) CFG-first hybrid. We report FID (↓) across different number of inference steps. Note that the additional NFE for obtaining the low-quality denoising term has been included. Blur is the searched degradation method.

| | NFE | | | | | |
|---|---|---|---|---|---|---|
| METHOD | 8 | 10 | 20 | 40 | 100 | 200 |
| DBIM (ZHENG ET AL., 2025) | 7.59 | 4.54 | 4.14 | 4.05 | 3.94 | 3.94 |
| DBIM+CFG (HO & SALIMANS, 2021; ZHENG ET AL., 2025) | 5.10 | 4.34 | 3.69 | 3.54 | 3.49 | 3.48 |
| DBIM+1/2FMPG+1/2CFG | 5.98 | 5.41 | 3.80 | 3.77 | - | - |
| **DBIM+1/2CFG+1/2FMPG** | **4.35** | **3.86** | **3.53** | **3.48** | **3.47** | **3.47** |

recently proposed fast samplers for bridge models, namely DBIM (Zheng et al., 2025) and ECSI (Zhang et al., 2025b). On both the Edges→Handbags and DIODE benchmarks, we can achieve a substantial 2× acceleration compared to the vanilla DBIM baseline, consistently delivering superior restoration quality at lower NFE regimes.

Furthermore, the incorporation of frequency modulation into PG, which aligns the mechanism of prior exploitation in bridge models, yields substantial internal improvements. By tailoring the guidance scales to specific frequency bands, FMPG is fundamentally more efficient than uniform pixel-level guidance. This significant efficiency gain is robust and consistently observed across different datasets, as evidenced in Table 4 and Table 5.

**Integration with CFG.** We evaluate our combination with CFG, namely CFG-FMPG, on ImageNet. For the challenging large-scale ImageNet in-painting task, we introduce the cascaded CFG-FMPG strategy. We report FID, IS, and CA in Table 6 and Table 7, respectively. As shown, this strategy achieves an unprecedented 20× overall speedup compared to the current state-of-the-art DBIM sampler. Notably, CFG-FMPG attains an FID of 3.86 at only 10 NFE, significantly outperforming the baseline which requires significantly higher computational cost to reach comparable performance.

### 4.3. Ablation Study

**Analysis of Frequency Modulation.** We further observe that decoupled strategies, such as fixing HF to a constant while applying an inverted U-shape to LF (or vice versa), also yield effective improvements, with the performance gain varying according to the peak magnitude of the curve. Detailed quantitative results and parameter configurations for the Edges→Handbags and DIODE datasets are provided in Appendix D.2. Regarding the curve design, as illustrated in Figure 2, the mechanism proves highly robust: it does not necessitate meticulous engineering. A generic convex function that ascends or descends to a peak plateau at a specific timestep is sufficient to achieve significant gains.

**Integration with CFG.** To incorporate CFG (Ho & Salimans, 2021), we employ a lightweight fine-tuning strategy

*Table 7.* Quantitative results on the ImageNet (256 × 256) task. Methods are grouped by NFE (40, 20, 10). IS (↑) and CA (↑) denote Inception Score and Classification Accuracy, respectively. Blur is the searched degradation method.

| Method | ImageNet (256 × 256) | | |
|---|---|---|---|
| | NFE | IS ↑ | CA ↑ |
| DBIM (Zheng et al., 2025) | | 137.6 | 71.6 |
| DBIM+CFG | 10 | 149.4 | 74.5 |
| DBIM+**CFG-FMPG (OURS)** | | **154.4** | **75.1** |
| DBIM (Zheng et al., 2025) | | 141.7 | 72.2 |
| DBIM+CFG | 20 | 158.1 | 76.0 |
| DBIM+**CFG-FMPG (OURS)** | | **160.1** | **76.1** |
| DBIM (Zheng et al., 2025) | | 143.3 | 72.5 |
| DBIM+CFG | 40 | 159.1 | **75.8** |
| DBIM+**CFG-FMPG (OURS)** | | **159.3** | 75.6 |

for only 10,000 steps (negligible cost). To ensure fairness, we compare against the optimal CFG hyperparameters identified via an extensive grid search detailed in Appendix E. Furthermore, in Table 6, a reversed (FMPG-first) schedule performed worse than standalone CFG, validating our analysis regarding the distinct roles of semantic establishment and details refinement.

## 5. Related Work

**Guidance methods.** Existing guidance methods for diffusion models mainly explore two directions. One line, exemplified by CFG (Ho & Salimans, 2021), focuses on better exploiting condition information to improve condition alignment. Subsequent works such as NAG (Chen et al., 2025), DCFG (Anonymous, 2026), and FDG (Sadat et al., 2025) further refine this paradigm by making use of condition information, *e.g.* text, more stable and effective across different layers, timesteps, or frequency bands. Another line, exemplified by AG (Karras et al., 2024), improves the generation quality by constructing degraded predictions to provide a corrective guidance signal. Related methods such as SAG (Hong et al., 2023), PAG (Ahn et al., 2025), and SEG (Hong, 2024) similarly construct negative branches by perturbing the model's internal attention behavior or weakening its effective capacity, and then use the resulting contrast to steer sampling toward higher-quality regions. In comparison, our PG methods target a bridge-specific dimen-

sion that is absent in standard diffusion models: the strong exploitation of an informative prior. A more comprehensive discussion of these related works is provided in Appendix A.

**Bridge models.** Recent efforts have explored the generative dynamics (Liu et al., 2023a; Zhou et al., 2024; Zhang et al., 2025b), parameterization method (Zhou et al., 2024; Chen et al., 2023), prior representation (Wang et al., 2025b; Bolton et al., 2026), data space (Li et al., 2025b; Zhang et al., 2025a), and sampling methods (Zheng et al., 2025; Zhang et al., 2025b) of tractable bridge models. In this work, GuidedBridge makes the first attempt to improve bridge generation quality with a guidance method that further emphasizes their advantage, namely prior exploitation, which is orthogonal to most previous methods.

## 6. Conclusion

In this work, we propose PG and its extension FMPG, two training-free guidance methods tailored for diffusion bridge models that unlock the untapped potential of the source prior as an informative guidance signal. By analyzing the underlying physics of the bridge generative process, FMPG dynamically adapts the guidance scale to match the inherent mechanism of prior exploitation across sampling steps. Our framework is orthogonal to existing guidance methods for condition alignment and integrates seamlessly with methods such as CFG, enabling cumulative gains through complementary mechanisms, *i.e.*, CFG-FMPG. Extensive experiments validate that our guidance methods achieve superior generation quality without sacrificing sampling efficiency. We hope that this work inspires further exploration into the structural properties of bridge priors and physics-informed guidance mechanisms.

## Acknowledgement

This work is supported by the National Natural Science Foundation of China (62550004, U24A20342, U25B6003, 92570001). The authors sincerely thank Yuji Wang at Tsinghua University, China and Chang Li at University of Science and Technology of China for the insightful discussions and suggestions.

## Impact Statement

This work accelerates Diffusion Bridge Models (DBMs) for image restoration and translation.

**Positive Societal Impacts:** Our method supports *Green AI* by significantly reducing inference energy and FLOPs, facilitating edge deployment and lowering carbon footprints. In restoration, it aids medical imaging (e.g., low-dose scans) and cultural heritage preservation through high-fidelity recovery.

**Potential Negative Societal Impacts:** Enhanced generation quality and speed inherently increase the risk of misuse, such as creating Deepfakes or generating misleading visual content at scale.

**Mitigation and Responsibility:** We advocate for integrating digital watermarking and detection tools. Furthermore, as models may amplify training biases (e.g., racial or gender), practitioners must perform fairness audits before sensitive deployments.

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

# A. Extended Related Work

## A.1. Generative Sampling

**Guided Sampling.** Recent perspectives have conceptualized the sampling process of diffusion and bridge models as a search problem within a high-dimensional latent space. Standard diffusion models rely on the learned score function as a gradient field to navigate this space stochastically (Song et al., 2021b; Ho et al., 2020). However, in data-to-data translation tasks, the search space is constrained by the source domain. Our proposed Prior Guidance (PG) injects an explicit guidance term derived from the contrast between a clean and a corrupted prior into the optimization landscape. This guidance term effectively prunes the search space by penalizing trajectories that deviate towards the weak prior manifold, thereby accelerating convergence to the target distribution from the prior distribution.

**Training-Free Guidance of Distribution-Transforming Models.** Diffusion Bridge Models (Zhou et al., 2024; Liu et al., 2023a; Zheng et al., 2025; Models, 2024) and Flow Matching represent two prominent classes of distribution-transforming generative models. Currently, research in this domain primarily focuses on guiding Flow Matching via energy functions (Feng et al., 2025). However, Flow Matching typically relies on a simple Gaussian prior, and bridge models have shown distinctive advantages of data-to-data transformation, *i.e.*, prior exploitation.

**Guidance with A Weak Model.** Existing literature has extensively explored guidance using a weak model (Karras et al., 2024; Ho & Salimans, 2021; Hong et al., 2023). However, these studies are predominantly tailored to general diffusion models. As a result, they fail to exploit the unique structural properties of the diffusion bridge Model, leaving its potential for specialized guidance largely unexplored.

## A.2. Guidance Mechanisms in Diffusion Models

Recent advances in training-free guidance have primarily explored two popular design spaces: mining the external condition information $c$ to maximize alignment, and mining the inherent potential of the pre-trained model $\theta$ to enhance generation quality.

### A.2.1. CONDITION MINING

This stream is dedicated to mining the potential of external conditions (e.g., text prompts) to maximize alignment. It aims to suppress unwanted semantic attributes by explicitly manipulating the conditioning signal.

- **Normalized Attention Guidance (NAG)** (Chen et al., 2025) addresses the instability of Classifier-Free Guidance (CFG) (Ho & Salimans, 2021) in few-step sampling regimes . It observes that standard output-space extrapolation leads to signal saturation and artifacts when sampling steps are aggressive. NAG moves the guidance operation into the attention mechanism with an L1-based normalization stabilizer.

- **Foresight Guidance (FSG)** (Wang et al., 2025a) challenges the linear extrapolation assumption of standard CFG. It reframes the guidance process as a fixed-point iteration problem, aiming to identify a "Golden Path" where conditional and unconditional latents achieve consistency through iterative refinement.

### A.2.2. MODEL POTENTIAL MINING

This stream operates without modifying external conditions. Instead, it focuses on mining the potential of the pre-trained model $\theta$ itself by constructing an internal "adversarial" state to guide the generation.

- **Self-Attention Guidance (SAG)** (Hong et al., 2023) pioneers the training-free self-guidance paradigm by leveraging the rich structural information embedded in intermediate self-attention maps. It constructs a negative branch by applying Gaussian blur to the attended content, effectively suppressing high-frequency details and guiding the generation toward enhanced stability and coherence.

- **Perturbed-Attention Guidance (PAG)** (Ahn et al., 2025) identifies that the self-attention mechanism is crucial for establishing global structural coherence. By replacing the attention map with an identity matrix during the negative pass, PAG effectively severs the contextual dependencies between tokens.

- **Smoothed Energy Guidance (SEG)** (Hong, 2024) reinterprets the self-attention mechanism through the lens of Energy-Based Models (EBMs). SEG constructs a negative branch by blurring the Query ($Q$) projection, which effectively "smoothes" the energy curvature to mitigate saturation artifacts.

- **Entropy Rectifying Guidance (ERG)** (Ifriqi et al., 2025) takes an information-theoretic approach. It perturbs the attention distribution by increasing the softmax temperature ($\tau$), driving the attention mechanism towards a high-entropy state where the model fails to focus on relevant features.

- **Stochastic Self-Guidance ($S^2$-Guidance)** (Chen et al., 2026a) extends the self-guidance paradigm to Diffusion Transformers (DiTs). Unlike U-Net-based methods that mask attention, $S^2$-Guidance constructs a weak model by randomly dropping transformer blocks during the negative pass. This "perturbed pass" filters out high-frequency details, allowing the guidance signal to enhance texture quality.

A.2.3. DYNAMIC AND FREQUENCY-ADAPTIVE GUIDANCE

Beyond the construction of the guidance direction, we also acknowledge a series of works that innovate on the guidance *scale*. These methods, which dynamically modulate the guidance strength, have provided valuable inspiration for our work.

- **Dynamic Classifier-Free Guidance (DCFG)** (Papalampidi et al., 2026) reformulates the guidance scale selection as an online optimization problem. Instead of a fixed schedule, it introduces a quantitative feedback mechanism that evaluates sample quality at each timestep. By maximizing alignment while strictly penalizing deviations (e.g., saturation), DCFG determines the optimal scale adaptively based on the model's instantaneous capacity.

- **Frequency-Decoupled Guidance (FDG)** (Sadat et al., 2025) identifies a limitation in uniform scaling: low frequencies (structure) are prone to saturation, while high frequencies (detail) require stronger boosting. FDG decomposes the guidance signal in the spectral domain, applying lower scales to low-frequency components and higher scales to high-frequency components to achieve high-fidelity generation without compromising diversity.

## B. Details of Corruption Methods

In our PG framework, the construction of a weak prior, $\mathcal{H}(\boldsymbol{x}_t)$, is essential for providing the negative gradient signal. To systematically study this, we investigate the performance of different corruption operator $\mathcal{H}(\cdot)$. Specifically, given the current state $\boldsymbol{x}_t$, the negative prior is training-freely constructed. We instantiate $\mathcal{H}$ with four distinct primitives:

**1. Gaussian Noise.** We define the noise injection operator $\mathcal{H}_{\text{noise}}$ as:

$$\mathcal{H}_{\text{noise}}(\boldsymbol{x}_t; \sigma) := \boldsymbol{x}_t + \boldsymbol{\epsilon}, \quad \text{where } \boldsymbol{\epsilon} \sim \mathcal{N}(\boldsymbol{0}, \sigma^2 \boldsymbol{I}). \tag{14}$$

This operator acts as a high-entropy filter that disrupts the signal coherence. By injecting unstructured isotropic Gaussian noise, $\mathcal{H}_{\text{noise}}$ effectively masks high-frequency details without altering the global semantic layout. Empirically, this serves as a "noisy observation" prior, forcing the model to distinguish between the structured signal in $\boldsymbol{x}_t$ and the purely stochastic component in $\mathcal{H}_{\text{noise}}(\boldsymbol{x}_t)$.

**2. Gaussian Blur.** The blurring operator $\mathcal{H}_{\text{blur}}$ is formalized as a convolution with a Gaussian kernel $G_\sigma$:

$$\mathcal{H}_{\text{blur}}(\boldsymbol{x}_t; k) := \boldsymbol{x}_t * G_{k,\sigma}. \tag{15}$$

Physically, $\mathcal{H}_{\text{blur}}$ functions as a low-pass filter in the frequency domain. It aggressively suppresses high-frequency components, such as sharp edges, textures, and fine-grained details, while preserving the low-frequency structural approximations. Using this as a negative prior explicitly penalizes "over-smoothness," thereby encouraging the guidance mechanism to recover sharpness and high-frequency details.

**3. JPEG Artifacts.** We denote the compression operator as $\mathcal{H}_{\text{jpeg}}$, which involves a lossy encoding-decoding cycle:

$$\mathcal{H}_{\text{jpeg}}(\boldsymbol{x}_t; Q) := \text{Decode}(\text{Encode}(\boldsymbol{x}_t, Q)). \tag{16}$$

This operator simulates the quantization errors inherent in JPEG compression with a low Quality Factor ($Q$). Unlike Gaussian noise or blur, $\mathcal{H}_{\text{jpeg}}$ introduces specific structural degradations, including block-wise discontinuities ($8 \times 8$ blocking artifacts) and Gibbs ringing effects around edges. This guides the model to reject characteristic compression artifacts and improves robustness.

**4. Super-Resolution / Pooling (SR4x).** The resolution degradation operator $\mathcal{H}_{\text{sr}}$ is defined by a downsampling-upsampling coupled process:

$$\mathcal{H}_{\text{sr}}(\boldsymbol{x}_t; s) := \text{Upsample}_s(\text{AvgPool}_s(\boldsymbol{x}_t)). \tag{17}$$

With a scaling factor $s = 4$, this operator effectively removes all sub-pixel fine-grained information. The naive upsampling step (e.g., nearest-neighbor or bilinear) results in a "pixelated" or aliased image. By using this as a negative reference, the prior guidance is incentivized to reconstructs missing high-frequency details during the generation process.

*Table 8.* Quantitative Comparison of Corruption Methods (FID ↓). We report FID scores on **Edges→Handbags** (**10,000 sampled images**) across different inference budgets (NFE). Notably, all results in this table are obtained using PG alone, **without FMPG**, providing a fair comparison of the corruption primitives under a unified setting.

| | NFE | | | |
|---|---|---|---|---|
| **CORRUPTION METHOD** | **10** | **20** | **40** | **100** |
| **GAUSSIAN NOISE** | **3.75** | **3.42** | 3.33 | 3.20 |
| **GAUSSIAN BLUR** | 4.01 | 3.50 | **3.26** | **3.15** |
| JPEG ARTIFACTS | 3.64 | 3.80 | 3.48 | 3.35 |
| SR / POOLING (4X) | 4.72 | 3.98 | 3.56 | 3.42 |

*Table 9.* Quantitative Comparison of Corruption Methods on **DIODE** (FID ↓). We evaluate the corruption primitives on the DIODE dataset across different inference budgets (NFE). Notably, all results in this table are obtained using PG alone, **without FMPG**, providing a fair comparison of the corruption primitives under a unified setting.

| | NFE | | | |
|---|---|---|---|---|
| **CORRUPTION METHOD** | **10** | **20** | **40** | **100** |
| **GAUSSIAN NOISE** | **6.25** | **3.77** | 2.96 | 2.63 |
| **GAUSSIAN BLUR** | 7.33 | 3.89 | **2.64** | **2.06** |
| JPEG ARTIFACTS | 6.58 | 4.25 | 3.42 | 3.15 |
| SR / POOLING (4X) | 8.45 | 5.60 | 4.15 | 3.72 |

## C. Quantitative Analysis on Guidance Parameters

In this appendix, we provide a detailed hyperparameter sensitivity analysis on the **DIODE** dataset across four computational budgets: **10, 20, 40, and 100 NFE**. Throughout this analysis, we uniformly adopt adding noise as the degradation operator.

### C.1. Low-Budget Regime (10 NFE)

Under the low-NFE setting, strong guidance is crucial. The top two performing scales are $w = 38.0$ (FID 6.59) and $w = 39.0$ (FID 6.64).

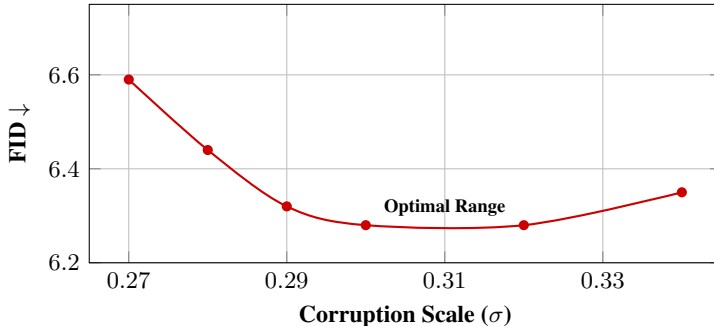

*Table 10.* **Guidance Scale** (10 NFE). Best at $w \in \{38.0, 39.0\}$.

| Scale ($w$) | FID | Scale ($w$) | FID |
|---|---|---|---|
| 26.0 | 9.18 | 36.0 | 6.73 |
| 28.0 | 8.62 | 37.0 | 6.65 |
| 30.0 | 8.12 | **38.0** | **6.59** |
| 32.0 | 7.80 | **39.0** | **6.64** |
| 34.0 | 7.50 | 41.0 | 6.93 |

*Figure 5.* **Corruption Scale (10 NFE).** Evaluated with $w = 38.0$. Optimal $\sigma \in [0.30, 0.32]$.

## C.2. Standard Regime (20 NFE)

With 20 NFE, the optimal guidance decreases. The best performance is achieved at $w = 20.0$ (FID 3.79) and $w = 21.0$ (FID 3.81).

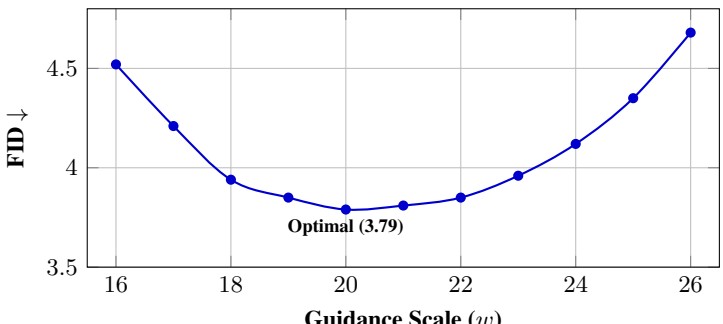

*Table 11.* **Guidance** (20 NFE). Best: $w \in \{20.0, 21.0\}$.

| Scale ($w$) | FID | Scale ($w$) | FID |
|---|---|---|---|
| 16.0 | 4.52 | **21.0** | **3.81** |
| 18.0 | 3.94 | 22.0 | 3.85 |
| 19.0 | 3.85 | 24.0 | 4.12 |
| **20.0** | **3.79** | 26.0 | 4.68 |

*Figure 6.* **Guidance Scale (20 NFE).** Convex shape confirms optimality at $w = 20.0$.

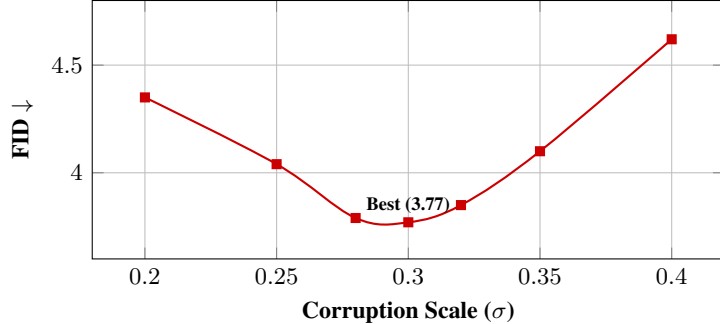

*Table 12.* **Corruption** (20 NFE). Best: $\sigma = 0.30$.

| Corr. ($\sigma$) | FID ↓ |
|---|---|
| 0.20 | 4.35 |
| 0.25 | 4.04 |
| **0.30** | **3.77** |
| 0.35 | 4.10 |
| 0.40 | 4.62 |

*Figure 7.* **Corruption Scale (20 NFE).** Robust around $\sigma = 0.30$.

## C.3. High-Fidelity Regime (40 NFE)

Optimal guidance scale shifts to $w \approx 14.0$. The top two scales are $w = 14.0$ (FID 2.96) and $w = 13.0$ (FID 3.05).

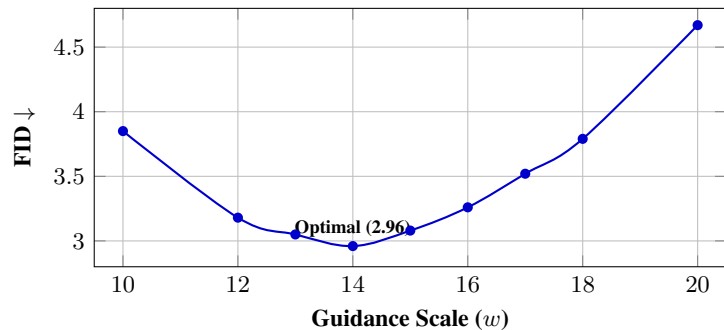

*Table 13.* **Guidance** (40 NFE). Best: $w \in \{13.0, 14.0\}$.

| SCALE ($w$) | FID | SCALE ($w$) | FID |
|---|---|---|---|
| 10.0 | 3.85 | 15.0 | 3.08 |
| 12.0 | 3.18 | 16.0 | 3.26 |
| **13.0** | **3.05** | 18.0 | 3.79 |
| **14.0** | **2.96** | 20.0 | 4.67 |

*Figure 8.* **Guidance Scale (40 NFE).** Shifted lower to $w = 14.0$.

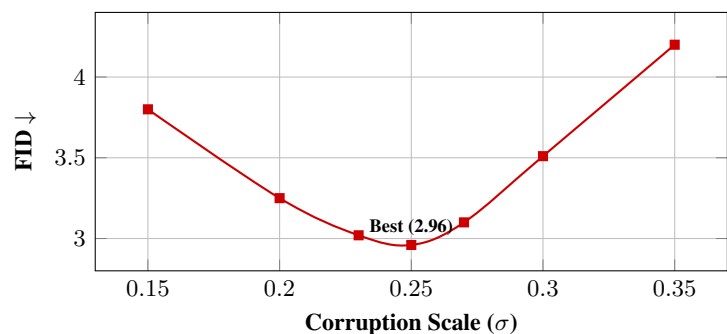

*Table 14.* **Corruption** (40 NFE). Best: $\sigma = 0.25$.

| CORR. ($\sigma$) | FID $\downarrow$ |
|---|---|
| 0.15 | 3.80 |
| 0.20 | 3.25 |
| **0.25** | **2.96** |
| 0.30 | 3.51 |
| 0.35 | 4.20 |

*Figure 9.* **Corruption Scale (40 NFE).** Best at clean prior $\sigma = 0.25$.

### C.4. Converged Regime (100 NFE)

At 100 NFE, best FID $\approx 2.67$. The optimal guidance scales are $w = 9.0$ (FID 2.67) and $w = 10.0$ (FID 2.77).

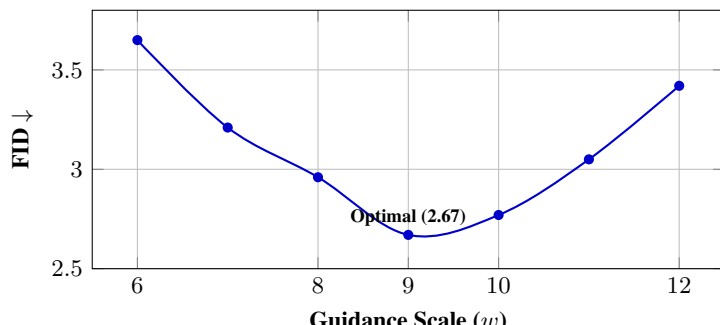

*Table 15.* **Guidance** (100 NFE). Best: $w \in \{9.0, 10.0\}$.

| SCALE ($w$) | FID $\downarrow$ |
|---|---|
| 6.0 | 3.65 |
| 8.0 | 2.96 |
| **9.0** | **2.67** |
| **10.0** | **2.77** |
| 12.0 | 3.42 |

*Figure 10.* **Guidance Scale (100 NFE).** Converged at $w = 9.0$.

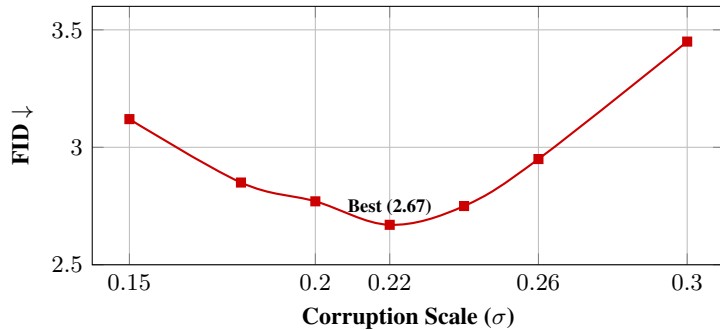

*Table 16.* **Corruption** (100 NFE). Best: $\sigma = 0.22$.

| CORR. ($\sigma$) | FID $\downarrow$ |
| --- | --- |
| 0.15 | 3.12 |
| 0.20 | 2.77 |
| **0.22** | **2.67** |
| 0.26 | 2.95 |
| 0.30 | 3.45 |

*Figure 11.* **Corruption Scale (100 NFE).** Very clean prior required ($\sigma = 0.22$).

# D. Frequency-Specific Guidance Analysis

To validate our Frequency-Modulated Prior Guidance (FMPG), we conducted an ablation study by restricting the guidance signal to specific frequency bands using FFT decomposition.

---

**Algorithm 1** Frequency-modulated Prior Guidance for Bridge Models

---

1: **Input:** Pre-trained denoiser $D_{\boldsymbol{\theta}}$, source image $\boldsymbol{x}_T$, steps $\{t_N, \ldots, t_0\}$, corruption $\mathcal{H}$, scales $w_{\text{low}}, w_{\text{high}}$.
2: **Output:** Sampled image $\boldsymbol{x}_0$.
3: $\boldsymbol{x} \leftarrow \boldsymbol{x}_T$
4: **for** $i = N$ **to** $1$ **do**
5: $\quad \boldsymbol{x}^{\text{bad}} \leftarrow \mathcal{H}(\boldsymbol{x})$ $\quad \triangleright$ Construct weak prior (bad state)
6: $\quad \hat{\boldsymbol{x}}_0^{\text{good}}, \hat{\boldsymbol{x}}_0^{\text{bad}} \leftarrow D_{\boldsymbol{\theta}}(\boldsymbol{x}, t_i), D_{\boldsymbol{\theta}}(\boldsymbol{x}^{\text{bad}}, t_i)$ $\quad \triangleright$ Predict $\boldsymbol{x}_0$ from both states
7: $\quad \boldsymbol{\Delta} \leftarrow \hat{\boldsymbol{x}}_0^{\text{good}} - \hat{\boldsymbol{x}}_0^{\text{bad}}$ $\quad \triangleright$ Calculate guidance residual
8: $\quad \mathcal{F}_{\boldsymbol{\Delta}} \leftarrow \text{FFT}(\boldsymbol{\Delta})$ $\quad \triangleright$ Transform to frequency domain
9: $\quad \boldsymbol{\Delta}_{\text{low}} \leftarrow \text{iFFT}(\text{LowPass}(\mathcal{F}_{\boldsymbol{\Delta}}))$ $\quad \triangleright$ Extract low-frequency component
10: $\quad \boldsymbol{\Delta}_{\text{high}} \leftarrow \boldsymbol{\Delta} - \boldsymbol{\Delta}_{\text{low}}$ $\quad \triangleright$ Extract high-frequency component
11: $\quad \boldsymbol{\Delta}_{\text{guided}} \leftarrow w_{\text{low}} \cdot \boldsymbol{\Delta}_{\text{low}} + w_{\text{high}} \cdot \boldsymbol{\Delta}_{\text{high}}$ $\quad \triangleright$ Apply frequency-specific scales
12: $\quad \hat{\boldsymbol{x}}_0^{\text{target}} \leftarrow \hat{\boldsymbol{x}}_0^{\text{bad}} + \boldsymbol{\Delta}_{\text{guided}}$ $\quad \triangleright$ Rectify prediction with guidance
13: $\quad \boldsymbol{x} \leftarrow \text{DBIMStep}(\boldsymbol{x}, \hat{\boldsymbol{x}}_0^{\text{target}}, \boldsymbol{x}_T, t_i, t_{i-1})$ $\quad \triangleright$ Update state with 5 params
14: **end for**
15: **return** $\boldsymbol{x}$

---

## D.1. Static vs. Dynamic Frequency Modulation

To isolate the contribution of our dynamic scheduling strategy, we compare our **FMPG** (Frequency-Modulated Prior Guidance) against a static baseline, denoted as **Static-PG**. In the Static-PG setting, the guidance scale is kept constant and identical for both high and low-frequency components throughout the sampling trajectory (i.e., $w_{\text{hf}}(t) = w_{\text{lf}}(t) = \lambda_{\text{const}}$). This baseline assumes that prior information is equally reliable across all frequencies and time steps.

However, our analysis reveals that this assumption is suboptimal for Bridge processes. As illustrated in the main text, the Signal-to-Noise Ratio (SNR) of a Bridge process follows a U-shaped curve (Zhou et al., 2024; Zheng et al., 2025), implying that the intermediate steps ($t \approx 0.5$) are dominated by noise, while the trajectory boundaries ($t \to 0$ and $t \to 1$) contain cleaner signal.

**1. The Failure of Static Guidance:** Static-PG forces a compromise. If the scale is set high to capture details, it amplifies noise in the intermediate steps, leading to high-frequency artifacts. If set low to ensure stability, it fails to sufficiently correct the structural alignment or refine fine textures at the endpoints.

**2. The Advantage of FMPG (Ours):** Our FMPG decouples the frequency bands to align with the physics of the Bridge process:

- **High-Frequency (HF) as a U-Shape:** We adopt a U-shaped schedule for HF components. Guidance is strengthened at

the endpoints ($t = 0$ and $t = 1$) to perform fine-grained texture refinement on the high-SNR data, but relaxed in the noisy intermediate region to prevent artifact amplification.

- **Low-Frequency (LF) as an Inverted U-Shape:** Conversely, LF components follow an inverted U-shaped schedule. We amplify LF guidance in the high-uncertainty intermediate region. This acts as a strong "structural anchor," ensuring the global layout remains consistent when the signal is weakest, before tapering off at the boundaries where the data is already structurally sound.

In this section, we investigate the sensitivity of the baseline **Static-PG** method to the guidance scale parameter $w$. We performed a fine-grained grid search on the *Edges→Handbags* dataset (Isola et al., 2017) to identify the optimal static scale.

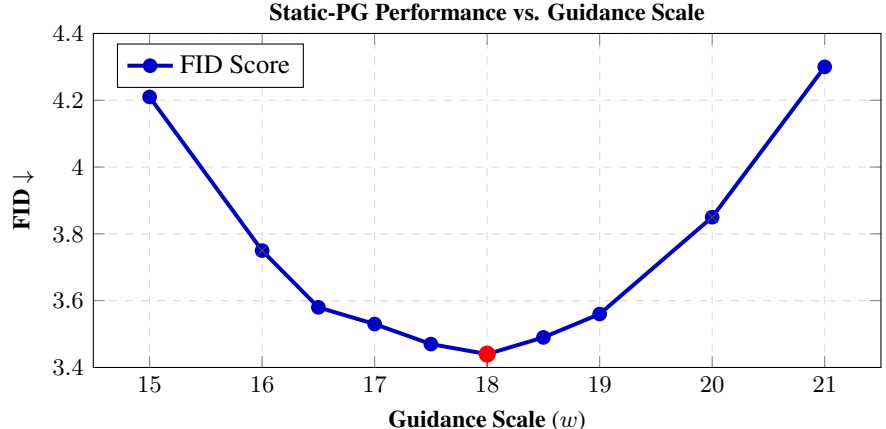

*Figure 12.* **Guidance Scale Search for Static-PG (Edges2Handbags** (Isola et al., 2017)). Ten discrete scale points are connected by line segments, forming a roughly U-shaped unimodal curve with the optimum at $w = 18$.

### D.2. Calibration of Dynamic FMPG Schedules

Building upon the optimal static baseline ($w_{\text{base}} = 18$), we further unlock the potential of our method by calibrating the dynamic modulation intensity. We independently fine-tune the modulation range for Low-Frequency (LF) and High-Frequency (HF) components to find the optimal deviation from the base scale.

**1. Low-Frequency (Structural) Calibration:** For the LF component, we employ an **Inverted U-shaped** schedule to strengthen structural guidance in the intermediate diffusion steps. We fix the endpoints at $w = 18$ and search for the optimal **Peak Scale** ($w_{\text{lf}}^{\text{max}}$) in the range of $[18.5, 20.5]$.

**2. High-Frequency (Texture) Calibration:** For the HF component, we employ a **U-shaped** schedule to relax guidance in the noisy intermediate steps. We fix the endpoints at $w = 18$ and search for the optimal **Minimum Scale** ($w_{\text{hf}}^{\text{min}}$) in the range of $[15.5, 17.5]$.

The ablation results are recorded in Table 17a and Table 17b respectively.

**Synergy of Optimal Schedules.** Finally, by simultaneously applying these optimized modulation strategies, specifically the **Inverted U-shape** for LF (peaking at $w_{\text{lf}}^{\text{max}} = 20.5$) and the **U-shape** for HF (dipping to $w_{\text{hf}}^{\text{min}} = 15.5$) as illustrated in the main text, our method achieves its peak performance reported in the repository: an FID of **3.19** (evaluated on 10,000 images) and **1.07** (evaluated on the full dataset of 138,567 images).

## E. Hyperparameter Tuning and Sensitivity Analysis

In this section, we provide detailed experimental evidence regarding the optimization of our baselines and the sensitivity of our proposed hyperparameters.

*Table 17.* **Ablation study on Dynamic Modulation Intensity.** The base scale is fixed at $w_{\text{base}} = 18$. We report FID scores (Heusel et al., 2017) (↓) for different modulation amplitudes.

*(a)* **Low-Frequency (LF) Tuning**. Increasing the peak of the Inverted U-shape.

| BASE SCALE | LF PEAK ($w_{\text{LF}}^{\text{MAX}}$) | FID |
|---|---|---|
| 18.0 | 18.5 | 3.42 |
| 18.0 | 19.0 | 3.37 |
| 18.0 | 19.5 | 3.35 |
| 18.0 | 20.0 | 3.30 |
| 18.0 | 20.5 | 3.28 |
| 18.0 | 21.0 | 3.31 |

*(b)* **High-Frequency (HF) Tuning**. Decreasing the trough of the U-shape.

| BASE SCALE | HF MIN ($w_{\text{HF}}^{\text{MIN}}$) | FID |
|---|---|---|
| 18.0 | 17.5 | 3.43 |
| 18.0 | 17.0 | 3.37 |
| 18.0 | 16.5 | 3.35 |
| 18.0 | 16.0 | 3.30 |
| 18.0 | 15.5 | 3.26 |
| 18.0 | 15.0 | 3.28 |

### E.1. Optimization of DBIM+CFG Baseline on ImageNet

We first disclose the tuning process for the strong baseline: **DBIM** (Zheng et al., 2025) combined with Classifier-Free Guidance (DBIM+CFG) (Ho & Salimans, 2021) on ImageNet (Deng et al., 2009) ($256 \times 256$). To ensure a rigorous comparison, we exhaustively searched for the optimal scale $w$ at each computational budget (10, 20, and 40 NFE).

*Table 18.* **10 NFE (ImageNet).** Best at $w = 2.0$.

| SCALE ($w$) | FID ↓ |
|---|---|
| 1.5 | 4.93 |
| 1.7 | 4.66 |
| **2.0** | **4.33** |
| 2.3 | 4.37 |
| 2.6 | 4.42 |
| 3.0 | 8.24 |

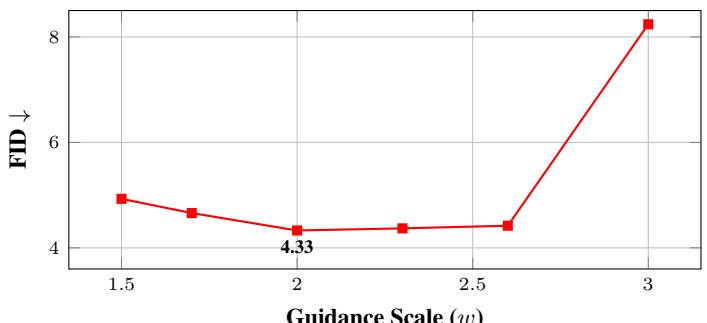

*Figure 13.* **10 NFE Baseline.** Optimal $w = 2.0$.

*Table 19.* **20 NFE (ImageNet).** Best at $w = 1.5$.

| SCALE ($w$) | FID ↓ |
|---|---|
| 1.2 | 3.72 |
| 1.4 | 3.69 |
| **1.5** | **3.68** |
| 1.6 | 3.69 |
| 1.8 | 3.71 |
| 2.0 | 3.72 |
| 2.5 | 3.77 |

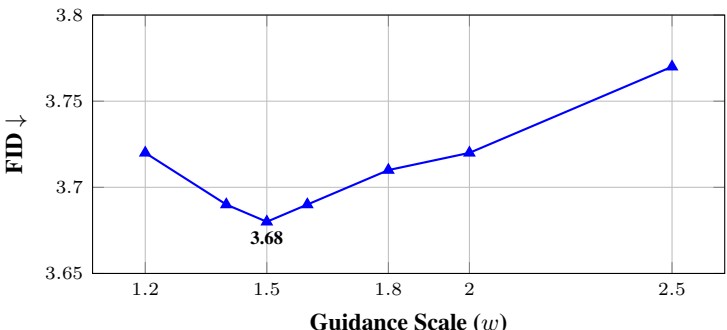

*Figure 14.* **20 NFE Baseline.** Optimal $w = 1.5$.

_Table 20._ **40 NFE (ImageNet).** Best at $w = 1.0$–1.1.

| SCALE ($w$) | FID (HEUSEL ET AL., 2017) $\downarrow$ |
|---|---|
| 0.7 | 3.57 |
| 0.9 | 3.55 |
| **1.0** | **3.54** |
| **1.1** | **3.54** |
| 1.4 | 3.55 |
| 1.5 | 3.57 |

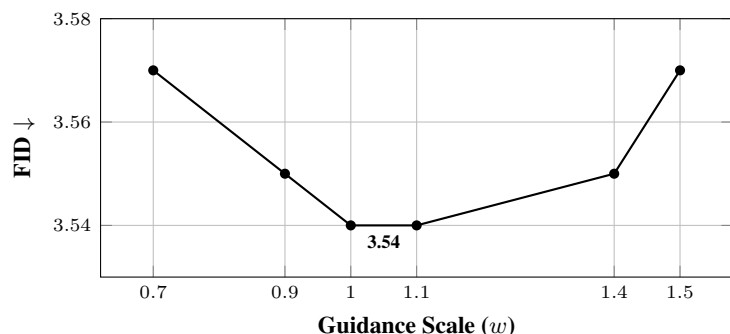

_Figure 15._ **40 NFE Baseline.** Optimal $w \approx 1.0$.

## E.2. Sensitivity Analysis of FMPG Parameters

We conducted fine-grained ablation studies on the **Start Ratio** ($\tau_{start}$) to identify the optimal configuration for Frequency-Modulated Prior Guidance (FMPG) under different computational budgets.

**Analysis at NFE = 10.** For the low-budget regime, we fixed the hyperparameters as follows:

- **CFG** (Ho & Salimans, 2021) Guidance Scale ($w$): 3.0
- **FMPG Low-Frequency Scale:** 1.45
- **FMPG High-Frequency Scale:** 1.35

Table 21 reports the FID scores (Heusel et al., 2017). The model achieves the best performance (FID = **3.86**) when frequency modulation starts at ratio **0.4**.

_Table 21._ **Ablation on FMPG Start Ratio (NFE = 10).** The nominal 10-NFE budget corresponds to 5 sampling steps under two-branch guidance. Since the first transition from $t = 1.0$ to $t = 0.9999$ is a booting step, the remaining four steps are effective denoising steps. We therefore keep only the discrete effective switching points present in our sweep. Fixed scales: CFG=2.0, Low=1.45, High=1.35.

| START RATIO ($\tau$) | 0.0 | 0.2 | **0.4** | 0.6 | 0.8 | 1.0 |
|---|---|---|---|---|---|---|
| FID (HEUSEL ET AL., 2017) ($\downarrow$) | 4.33 | 3.89 | **3.86** | 4.05 | 4.51 | 5.08 |

**Analysis at NFE = 20.** For the standard budget regime, the hyperparameters were set to:

- **CFG** (Ho & Salimans, 2021) Guidance Scale ($w$): 2.5
- **FMPG Low-Frequency Scale:** 1.2
- **FMPG High-Frequency Scale:** 1.11

Table 22 presents the results. The optimal sweet spot shifts slightly, achieving the best FID (Heusel et al., 2017) (**3.53**) at start ratio **0.3**.

_Table 22._ **Ablation on FMPG Start Ratio (NFE = 20).** Fixed scales: CFG=2.5, Low=1.2, High=1.11. Optimal start ratio is 0.3.

| START RATIO ($\tau$) | 0.0 | 0.2 | **0.3** | 0.4 | 0.5 | 0.6 | 0.7 | 1.0 |
|---|---|---|---|---|---|---|---|---|
| FID (HEUSEL ET AL., 2017) ($\downarrow$) | 3.69 | 3.55 | **3.53** | 3.54 | 3.55 | 3.58 | 3.61 | 3.74 |

**Analysis at NFE = 40.** For the high-fidelity regime, the hyperparameters were configured as:

- **CFG** (Ho & Salimans, 2021) Guidance Scale ($w$): 2.1

- **FMPG Low-Frequency Scale:** 1.1
- **FMPG High-Frequency Scale:** 1.04

Table 23 summarizes the FID scores (Heusel et al., 2017). Similar to the low-budget regime, we observe an optimal start ratio at **0.4** with an FID of **3.47**.

*Table 23.* **Ablation on FMPG Start Ratio (NFE = 40).** Fixed scales: CFG=2.1, Low=1.1, High=1.04. Optimal start ratio is 0.4.

| START RATIO ($\tau$) | 0.0 | 0.1 | 0.2 | 0.3 | **0.4** | 0.5 | 0.6 | 0.8 | 0.9 | 1.0 |
|---|---|---|---|---|---|---|---|---|---|---|
| FID (HEUSEL ET AL., 2017) ($\downarrow$) | 3.54 | 3.53 | 3.51 | 3.49 | **3.47** | 3.50 | 3.58 | 3.75 | 3.82 | 3.94 |

# F. Qualitative Results

We provide additional visual results to demonstrate the effectiveness of our method across different domains: DIODE (outdoor scenes), Edges2Handbags (edge-guided generation), and ImageNet (class-conditional in-painting). For all experiments, we consistently use the pre-trained checkpoints provided by DBIM (Zheng et al., 2025) and a first-order ODE sampler.

## F.1. DIODE: Generation Quality across Different NFEs

We visualize the evolution of generation quality on the DIODE dataset.

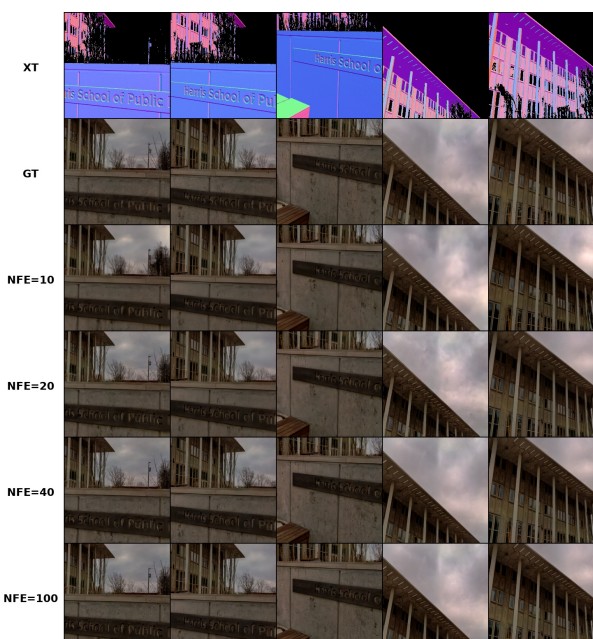 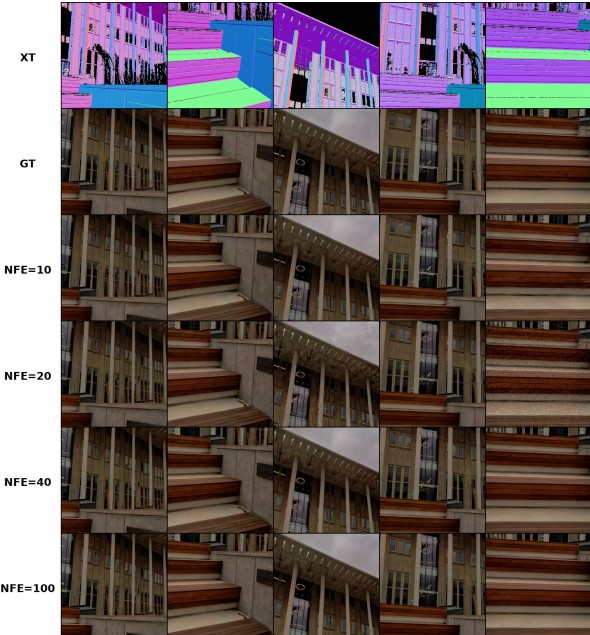

*Figure 16.* **Qualitative Results on DIODE (Part I).** Visual comparison of the first two samples.

## F.2. Edges2Handbags: Texture Evolution

We further demonstrate robustness on the Edges2Handbags dataset.

## F.3. ImageNet: High-Fidelity Synthesis

Finally, we present results on the challenging ImageNet ($256 \times 256$) dataset.

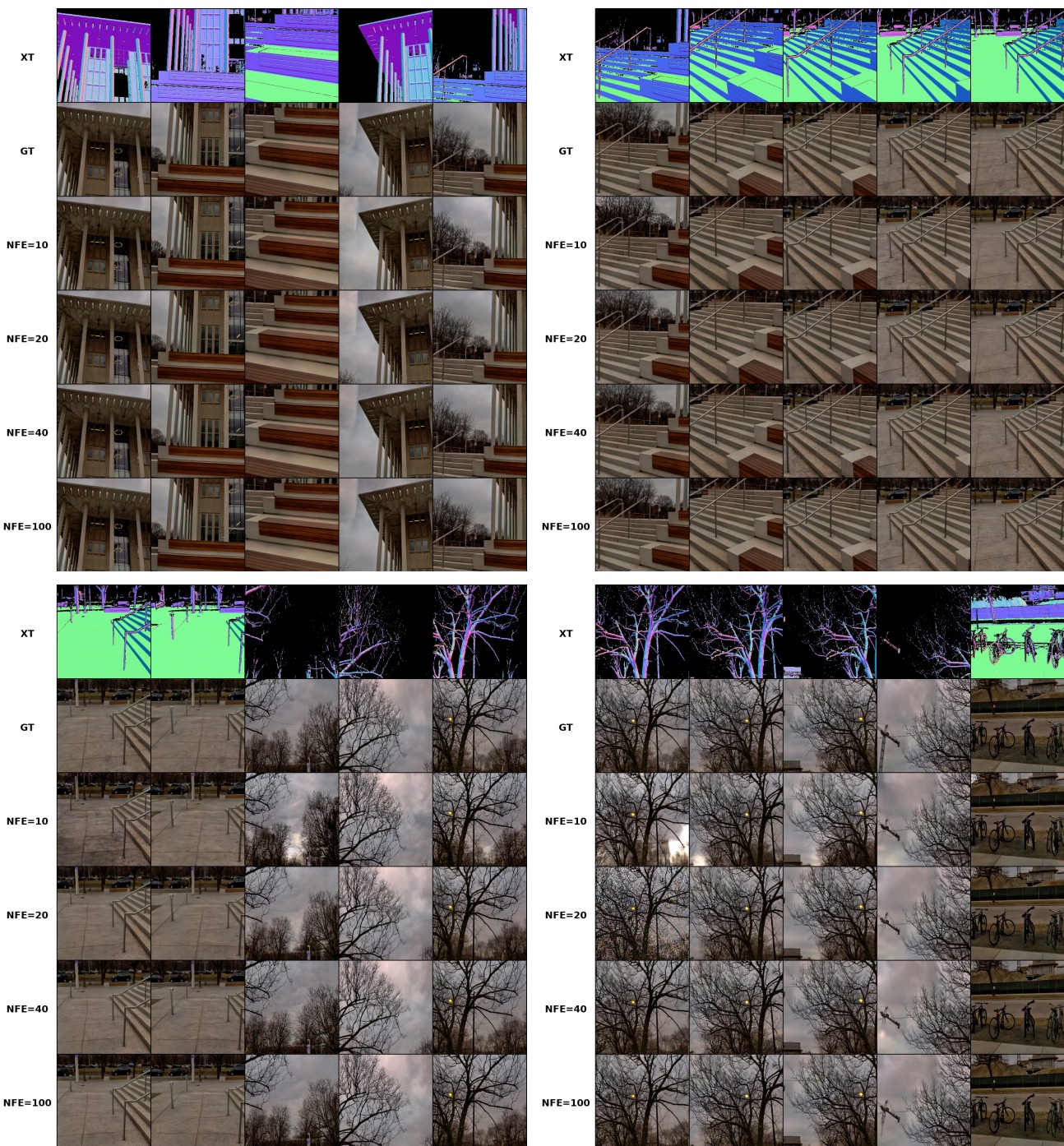

*Figure 17.* **Qualitative Results on DIODE (Part II).** Visual comparison of additional samples.

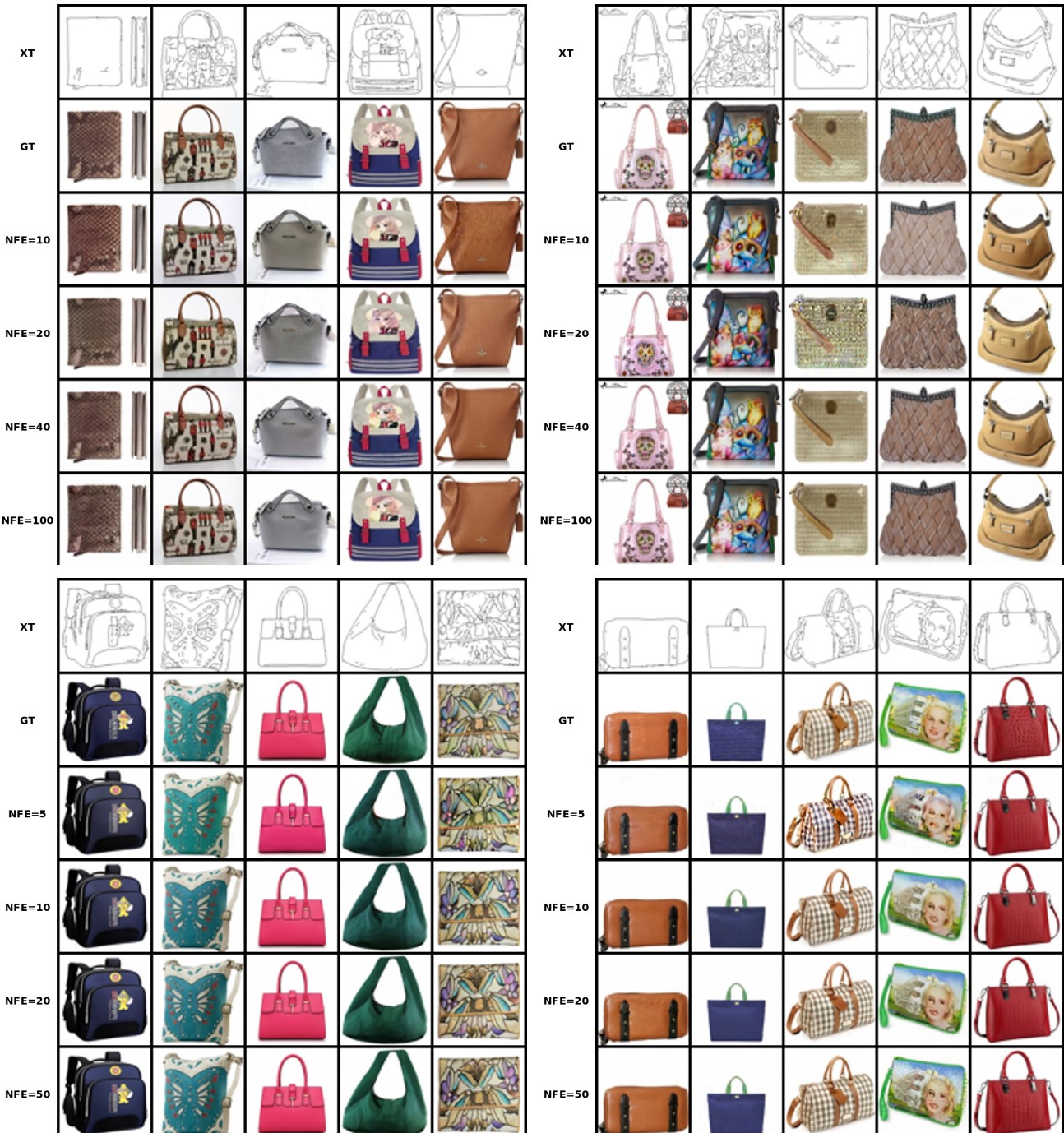

*Figure 18.* **Qualitative Results on Edges2Handbags (Part I).** Visual comparison of additional samples.

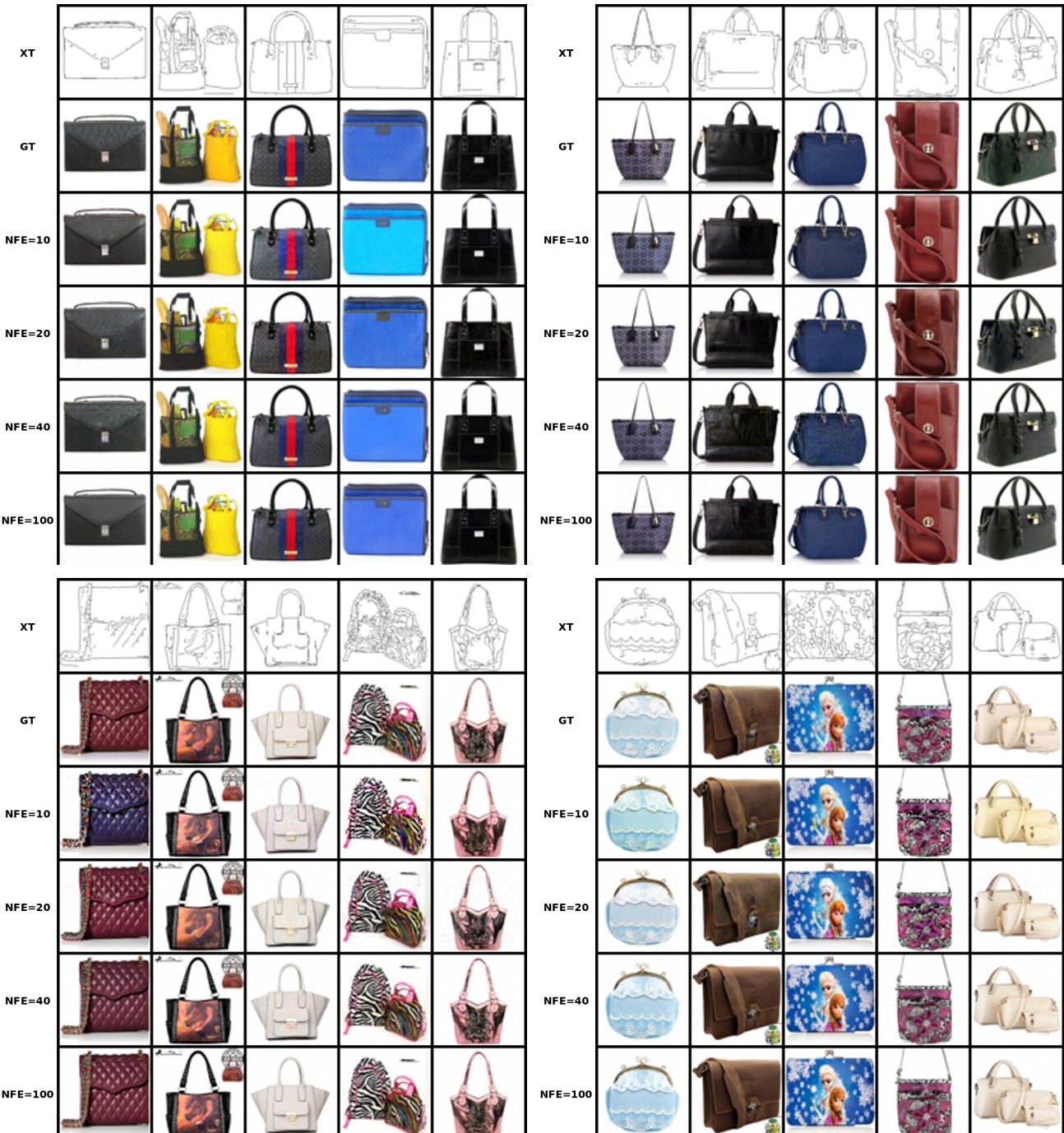

*Figure 19.* **Qualitative Results on Edges2Handbags (Part II).** Visual comparison of additional samples.

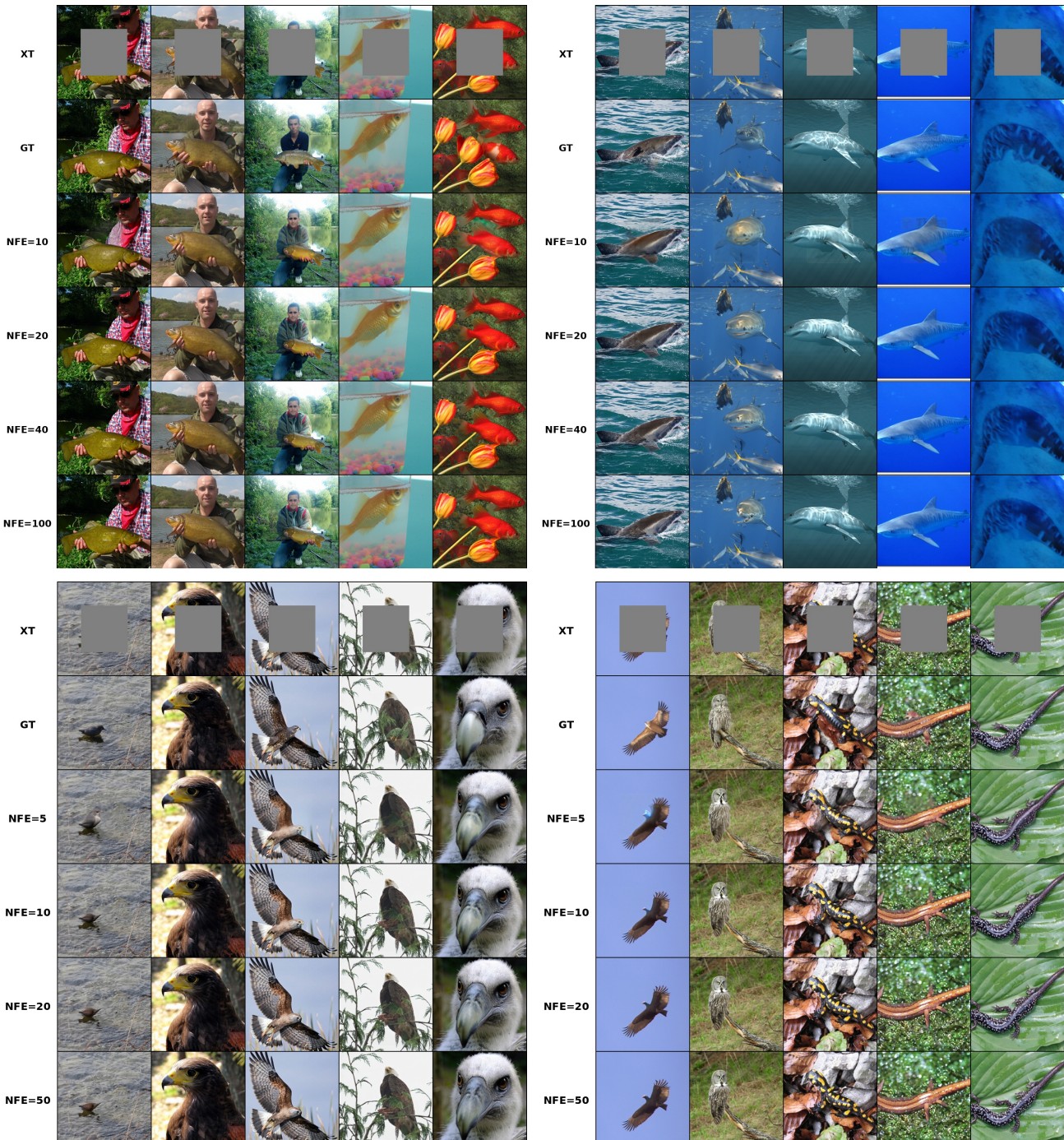

*Figure 20.* **Qualitative Results on ImageNet (Part I).** Visual comparison of additional samples.

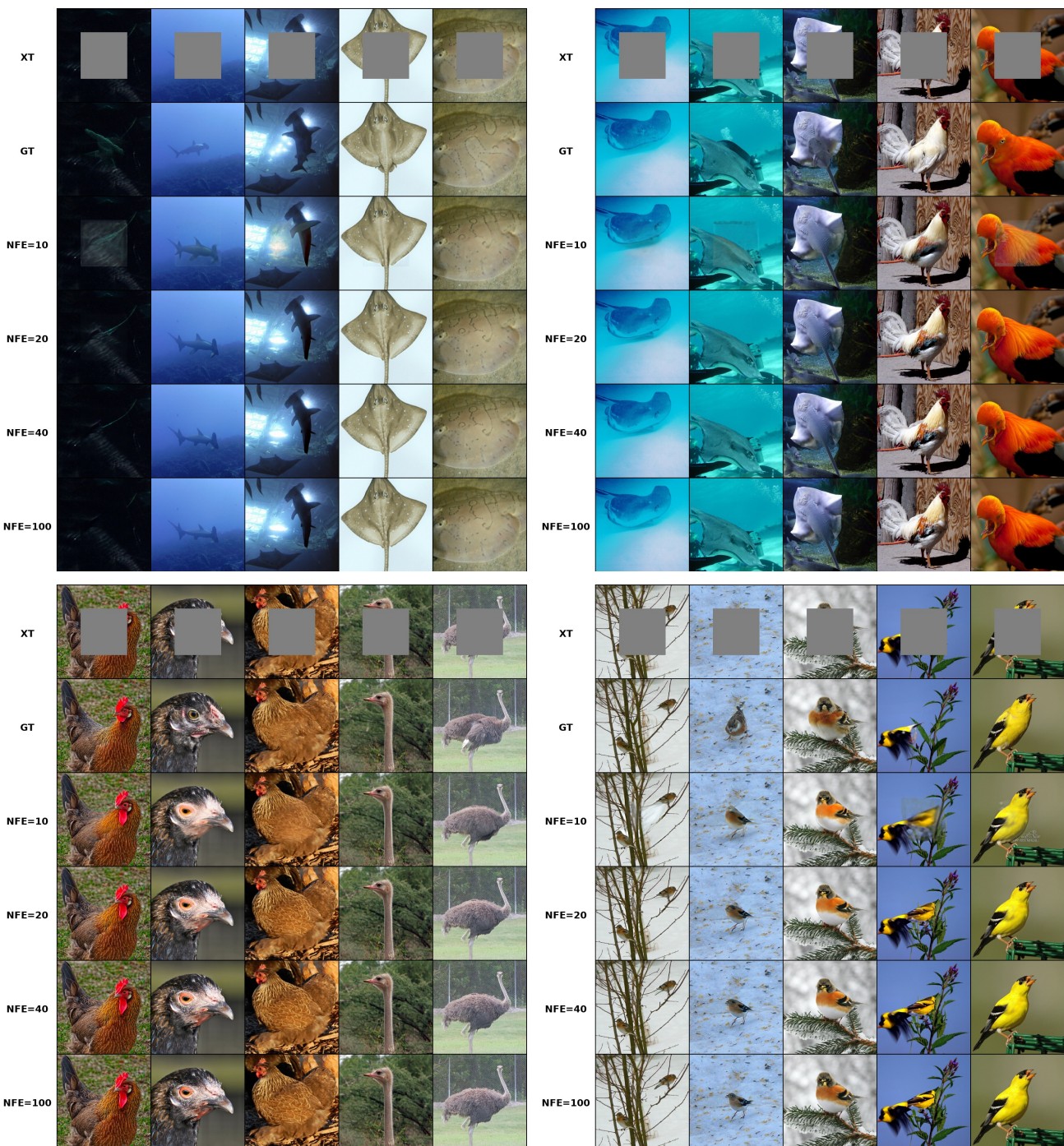

*Figure 21.* **Qualitative Results on ImageNet (Part II).** Visual comparison of additional samples.

