# OpenReview forum: "GuidedBridge: Training-freely Improving Bridge Models with Prior Guidance"
_ICML.cc/2026/Conference — ICML 2026 regular_

### Official Review · Reviewer_3qer · 2026-03-04

**Soundness:** 4
**Presentation:** 3
**Significance:** 3
**Originality:** 4
**Overall Recommendation:** 3
**Confidence:** 4

**Summary:**

This paper proposes Prior Guidance (PG) for data-to-data diffusion bridge models, inspired by Classifier-Free Guidance (CFG) and Auto-Guidance (AG) for noise-to-data diffusion models. PG constructs a weak prior to create a low-quality denoising result and forces the model to better exploit the informative prior information through extrapolating between the results from the original strong prior and the constructed weak prior. The authors propose Frequency-Modulated Prior Guidance (FMPG) that dynamically adjusts guidance scales for low-frequency (LF) and high-frequency (HF) components to align with the bridge's generative dynamics. For tasks with weak priors (e.g., inpainting), they introduce a cascaded CFG-FMPG framework that combines CFG with FMPG.

The key contributions can be summarized as follows:
(1) A novel, training-free guidance paradigm PG specifically designed for bridge models
(2) FMPG mechanism that tailors guidance scales to frequency bands based on the U-shaped SNR evolution of bridge models
(3) A hybrid CFG-FMPG that integrates CFG and FMPG to handle tasks where the initial prior is insufficient
(4) Experimental results demonstrate consistent improvements over state-of-the-art bridge models (DDBM, DBIM) across multiple datasets

**Compliance With Llm Reviewing Policy:**

Affirmed.

**Key Questions For Authors:**

Q1. In section 5.2, the authors reported 2x acceleration over DBIM and 1.5x speedup with FMPG. However, there is no running time provided. Does this acceleration mean that FMPG requires less sampling step (NFE) to achieve certain FID metrics? If so, the author should clarify this in main content. Meanwhile, it is suggested the authors provide the implementation details, e.g. GPU device, fine-tuning configuration etc.

Q2. As described in weakenesses 2, how about the performance of PG on modalities other than image, e.g. text, audio?

Q3. Table 7 and 8 show variations in performance of different degradation operators. (e.g., Noise works best for low NFE, Blur for high NFE). Are there any explanations about this variation? Is there a theoretical basis or a simple heuristic to automatically select the optimal degradation operator based on the task or the current sampling step?

Q4. Many modern diffusion/bridge models are distilled to very few steps (1-4 steps). How does PG perform when the base model is already heavily distilled?

**Limitations:**

Please refer to weakenesses and questions.

**Strengths And Weaknesses:**

# Strengths

1. A novel, training-free guidance paradigm PG specifically designed for diffusion bridge models and two variations FMPG and CFG-FMPG are proposed.
2. The method delivers substantial empirical benefits. The paper reports up to a 20x speedup in sampling steps while maintaining or improving image quality.

# Weakenesses

1. The core mechanism of Prior Guidance relies on the existence of an informative prior. In scenarios where the prior is extremely weak or missing, the weak prior construction yields diminishing returns. Although the authors propose a hybrid CFG solution, this adds complexity and requires a mechanism to switch between strategies or tune two sets of hyperparameters.
2. To verify the effectiveness of PG, the experiments mainly focus on image-based bridge models. Since CFG has been demonstrated to be easily generalized to different modalities, the performance of PG on other modalities, text or audio, is unclear.
3. There are several writing flaws in the manuscript.
(1) Section 4.4, line 263, the hyperref of Figure is missing.
    Section 5.1 line 319, no hyperref of Eqs 7 and 8.
    Section 5.2, line 368, the hyperref of Appendix 1c leads to Figure 1 not appendix.
    Table 2, DDBM and DBIM refer to the same paper.
    etc.
(2) Abbreviations and their full forms should be defined at first use. e.g. NFE.
It is suggested the authors check carefully about the manuscript to avoid similar writing flaws.

---

> ### Author Rebuttal · Authors · 2026-03-31
>
> **Response to Reviewer 3qer**
>
> ***
>
> **Response to W1: Existence of informative prior**
>
> We thank the reviewer for the insightful comment.
>
> (1) On informative priors
>
> Reliance on an informative prior is intrinsic to bridge models. Strong priors enable the superior performance of bridge models, while extremely weak priors inherently limit them. PG is designed to better exploit prior information, consistently improving performance in a training-free manner **across both strong and weak prior settings**.
>
> (2) On the hybrid CFG + PG strategy
>
> For weak priors, we develop a simple cascaded CFG then PG strategy to strengthen bridge models. Each step uses only one guidance mechanism, **without mixing guidance methods (e.g., parallel or nested combinations)**. The method remains simple and yields empirical gains.
>
> ***
>
> **Response to W2 and Q2: Performance on other modalities**
>
> We thank the reviewer for this helpful comment.
>
> (1) On extending to additional modalities
>
> We agree that evaluating PG on a broader range of modalities would provide a more comprehensive validation of its effectiveness. To this end, we have further extended our experiments beyond image-based settings to include **audio super-resolution, image-to-video generation tasks, and high-resolution image editing tasks**. The results show that PG consistently improves performance across these modalities, demonstrating its general applicability beyond images. Due to character limitations, please kindly visit our response to Reviewer 4RV4 and YyvJ.
>
> (2) On text-based bridge models.
> Regarding text modality, we note that bridge-based generative models for text are still relatively limited. To the best of our knowledge, we have not identified established text-based bridge model benchmarks that would allow for a meaningful evaluation of PG in this setting. We believe this is an interesting direction for future work and will clarify this point in the revision.
>
> ***
>
> **Response to W3: Writing flaws**
>
> We sincerely thank the reviewer for the careful reading and for pointing out these issues. We agree that these writing flaws should be corrected. In the revision, we will fix all the mentioned problems and thoroughly check the manuscript to ensure overall clarity and correctness.
>
> ***
>
> **Response to Q1: Sampling efficiency**
>
> We thank the reviewer for raising this important point. Here, “speedup” refers to requiring fewer NFEs to reach comparable generation quality. In our setup, both PG/FMPG inference and CFG fine-tuning are run on 4× RTX 4090 GPUs, and the CFG fine-tuning uses 10,000 training steps.
>
> The 2× speedup over DBIM is directly supported by our results. For example, in Table 4, DBIM+FMPG(Noise) at NFE = 20 already achieves a better result (3.20) than DBIM at NFE = 40 (3.35).
>
> FMPG is also clearly more efficient than PG. A particularly strong example appears in Table 4: DBIM+FMPG(Noise) reaches 2.62 at NFE = 40, while DBIM+PG(Noise) needs NFE = 100 to reach a comparable 2.63. This indicates an efficiency gap of up to about 2× in this setting.
>
> ***
>
> **Response to Q3: Performance of degradation operators and explanations**
>
> We thank the reviewer for this insightful question. A key reason is that when the number of sampling steps is large, there are also more steps near the terminal stage, where the intrinsic bridge noise is already very small. Under full-trajectory PG with a fixed degradation strength, applying constant additive noise at these late-stage steps can repeatedly push the degraded branch out of distribution. This may explain why blur performs better than noise in Tables 7 and 8 at larger step counts.
>
> In the more flexible FMPG setting, noise-based degradation still performs better in most settings. For other tasks, we still rely on empirical search to choose a suitable degradation operator; this is also the case for Autoguidance, which likewise requires searching for an appropriate degradation design.
>
> ***
>
> **Response to Q4: PG for distilled bridge models**
>
> We thank the reviewer for this important question.
>
> (1) Heavily distilled models.
>
> While standard diffusion models have been distilled to 1–4 steps, distillation for bridge models remains underexplored, with only a few works (e.g., Inverse Bridge Matching Distillation and Consistency Diffusion Bridge Models) and no public implementations, limiting direct evaluation. The interaction between distillation and guidance (including PG) can be non-trivial and left for future study.
>
> (2) Few-step performance.
>
> To supplement additional evidence, we further evaluate PG under extremely low-step settings on DBIM. Due to a singularity at $t=T$, an initial noise step is required, so 3 and 4 NFE correspond to only 2 and 3 effective transitions. Even under such tight budgets, PG still brings significant improvements over the DBIM baseline.
>
> | NFE | Method | FID |
> | :--- | :--- | :--- |
> | 3 | DBIM| 6.484 |
> | 3 | DBIM+FMPG | **4.285** |
> | 4 | DBIM | 5.650 |
> | 4 | DBIM_FMPG | **4.044** |

---

> > ### Author Rebuttal · Reviewer_3qer · 2026-04-03
> >
> > Thanks for the response.
> >
> > I'm still curious about whether PG can be adapted to situations like noise-to-image with conditions of other modalities. If it can only be used for bridges with two same modalities, the application scenerios are too limited. And as the authors confirmed, the optimal choice of degradation operators requires experience. I prefer to keep my score.

---

> > > ### Author Response · Authors · 2026-04-04
> > >
> > > Thank you for your insightful comments and important questions. We provide our reply and new evidence below.
> > >
> > > ***
> > >
> > > **Response to Concern 1: Application Scenarios of PG**
> > >
> > >
> > > **PG can be adapted to noise-to-image generation with conditions of other modalities**
> > >
> > > In response to your concern, we have verified PG in text-to-image generation task with diffusion models.
> > > Strictly aligning with CFG++ (ICLR 2025) guidance baseline, we evaluate both standard (SD v1.5, NFE=10) and distilled (SDXL-Lightning, NFE=4) models. Under identical experimental settings, PG outperforms baselines across the majority of metrics in both settings.
> > >
> > > Table A: Quantitative comparison (10K images) with standard diffusion sampling (SD v1.5)
> > > | NFE | Method | CLIP ↑ | FID ↓ | IS ↑ |
> > > | :--- | :--- | :--- | :--- | :--- |
> > > | 10 | CFG | 0.3119 | 15.61 | 31.17 |
> > > | 10 | CFG++ | 0.3119 | 15.40 | 31.15 |
> > > | 10 | PG (Ours) | **0.3146** | **14.38** | **35.26** |
> > >
> > > Table B: Quantitative comparison (5K images) with distilled sampling (SDXL-Lightning)
> > > | NFE | Method | CLIP ↑ | FID ↓ | IS ↑ |
> > > | :--- | :--- | :--- | :--- | :--- |
> > > | 4 | CFG | 0.3131 | 25.74 | 36.45 |
> > > | 4 | CFG++ | **0.3181** | 25.28 | 36.48 |
> > > | 4 | PG (Ours) | 0.3144 | **22.19** | **36.62** |
> > >
> > > **Applications scenarios of PG are not too limited**
> > >
> > > Consider a target distribution $p(x_0)$ and a prior distribution $ p(x_T) $, a diffusion model can reconstruct $p(x_0)$ from Gaussian noise and a bridge model can reconstruct $p(x_0)$ from clean prior representation at $t=T$. We denote the pre-trained diffusion or bridge denoising network as $D_{\theta}$, other condition signal as $c$, and the degradation operator in PG as $\mathcal{H}$. We analyze the application scenarios of PG as follows.
> > >
> > > (1) **Bridge models: informative prior $x_T$ without other condition.** In this scenario, for image, we have improved image-to-image translation. For audio, we have improved audio-to-audio super-resolution backed by a scalable latent bridge.
> > >
> > > **Namely, in this scenario, bridge models have shown advantages over conditional diffusion baseline or counterpart in prior work. Our PG is the first guidance method to further improve them by emphasizing prior exploitation from $t=T$.**
> > >
> > > (2) **Conditional diffusion models: Gaussian prior with other condition $c$.** In this scenario, CFG has emphasized condition alignment by extrapolating $D_{\theta}(x_0 | x_t, t, c)$ and $D_{\theta}(x_0 | x_t, t)$.
> > > Our PG method can further enhance quality by exploiting $x_t$ at each sampling step, namely extrapolating $D_{\theta}(x_0 | x_t, t, c)$ and $D_{\theta}(x_0 | \mathcal{H}(x_t), t)$. For image, we have improved **text-to-image** generation for both pre-trained and distilled diffusion models.
> > >
> > > **Namely, in this scenario, CFG can strengthen alignment with condition $c$, and our PG further enhances guidance with the axis of prior exploitation, improving quality with the same NFE and guidance scale searched for CFG.**
> > >
> > > (3) **Conditional bridge models: informative prior $x_T$ with other condition $c$.** In this scenario, CFG only improves alignment with $c$. Our PG method supplements the guidance.
> > >
> > > For image, we have improved class-conditioned image-to-image in-painting and high-resolution image editing results. For video, we have improved text-conditioned **image-to-video** generation.
> > >
> > > **Namely, in this scenario, jointly considering CFG and PG fully leverages indicative signals, $c$ and $x_T$.**
> > >
> > > **We believe our theoretical analysis and experimental scope for PG methods have supported a wide range of application scenarios, rather than very limited scenarios.**
> > >
> > > ***
> > >
> > > **Response to Concern 2: Optimal Choice of Degradation Operators**
> > >
> > >
> > > **Design of PG.**
> > > It should be noted that PG does not require **any training design and conditioning technique**, relying solely on experience during **inference**.
> > >
> > > **Robustness of PG.**
> > > For tasks including image generation (SD1.5/SDXL-lightning, diffusion in latent space), in-painting (DBIM, bridge in data space), translation (DDBM and DBIM, bridge in data space), and high-resolution image editing (ViBT, bridge in latent space), simple noise addition has served as a universal degradation operator.
> > > For other data modalities, such as audio and video, it is natural to examine the optimal degradation operator.
> > >
> > > **Simplicity of PG.**
> > > We have presented a simple and consistent two-stage tuning strategy, where the second stage, namely searching guidance scale, is shared across guidance methods.
> > > In our extensive experiments, the first stage, namely selecting the optimal degradation operators **(mostly from adding noise, blurring, or pooling), can be efficiently performed with a small test batch and few inference steps, without any training**.
> > >
> > > **Our design and implementations will be fully open-sourced to facilitate future research.**
> > >
> > > We sincerely thank you for your careful review and valuable feedback.
> > > If your concerns are addressed, we would deeply appreciate your consideration in reassessing the score.

---

### Official Review · Reviewer_SLvh · 2026-03-07

**Soundness:** 2
**Presentation:** 2
**Significance:** 3
**Originality:** 2
**Overall Recommendation:** 4
**Confidence:** 3

**Summary:**

This paper proposes a simple and practical training-free guidance method for diffusion bridge models. This method is easy to implement and can be applied to existing pre-trained models without additional training. The idea of utilizing prior information in the bridge model is quite inspiring, and the proposed frequency modulation guidance method provides a reasonable extension to this. Experimental results on multiple image translation and restoration tasks demonstrate that this method significantly improves upon other excellent baseline methods.

**Compliance With Llm Reviewing Policy:**

Affirmed.

**Final Justification:**

The idea is interesting, despite limited theoretical contribution. The added experiments are convincing.

**Key Questions For Authors:**

Since the proposed method requires multiple denoiser evaluations at each timestep, have the authors considered leveraging predictions from previous timesteps to construct the guidance signal? For example, one could potentially use the difference between the current prediction and the previous-step prediction to reduce the number of denoiser calls.

**Limitations:**

yes

**Strengths And Weaknesses:**

# Strength

1. Simple and practical design. The proposed PG and FMPG methods are relatively simple and can be easily integrated into existing diffusion bridge models without modifying the training procedure.

2. Training-free and empirically effective. The approach operates at inference time and does not require additional training. Experimental results on several image translation and restoration tasks show consistent improvements over baseline bridge models.



# Weakness

1. The Prior Guidance (PG) proposed in this paper essentially follows the basic paradigm of existing guidance methods, namely, guiding the generation process by constructing two predictions of different qualities and performing extrapolation. Similar ideas have been extensively studied in existing work, such as classifier-free guidance, autoguidance, and a series of training-free auto-guidance. The main difference in this work is that it uses the difference between "good priors" and "degenerate priors" as the source of guidance, but the overall algorithm structure is quite similar to existing methods, thus its methodological innovation is relatively limited. For instance, diffusion bridge models translate from prior information $y$ to the posterior $p(x_0 \mid y)$, discussing how the dynamic changes to achieve $p(x_0 \mid y, c)$ would significantly improve the theoretical understanding of the proposed method.

2. The proposed method introduces several design choices and hyperparameters, including the guidance scale and the frequency-modulated schedules. These parameters appear to be chosen heuristically, and the paper provides limited theoretical guidance on how they should be selected. In particular, it is unclear whether the same settings would generalize to different diffusion bridge paths, or whether additional tuning would be required.

3. The paper contains several formatting issues (e.g., Line 263, Page 26, Figure 13, and Line 639), which slightly affect readability.

---

> ### Author Rebuttal · Authors · 2026-03-31
>
> **Response to Reviewer SLvh**
>
> ***
>
> **Response to W1: Methodological Innovation**
>
> We thank the reviewer for the insightful comments.
>
> (1) Difference from existing guidance methods
>
> While PG follows the general guidance paradigm by extrapolating between high and low quality denoising predictions, its key novelty lies in **expanding the guidance design space** by emphasizing prior exploitation of bridge models. To our knowledge, **PG is the first training-free guidance method tailored for bridge models**, improving generation quality and sampling efficiency even in few-step settings, and generalizing across multiple modalities and tasks, which constitutes a meaningful methodological contribution beyond existing guidance strategies.
>
> (2) On “how the dynamics change to achieve $p(x_0 | y, c)$
>
> We sincerely thank the reviewer for this insightful question. We may not have fully understood this question, and we would greatly appreciate further clarification if our interpretation does not address your concern. Assuming $c$ denotes an additional conditioning signal beyond the prior information $y$, we provide our understanding as follows.
>
> First, bridge models learn a score function $𝐷_{\theta}$ that models the transition from prior information $𝑦$ to target $x_0$, namely, $p(x_0 | y)$. In PG, we construct a degraded prior $\hat{y}$, which is not seen during training and thus leads to lower-quality denoising results. By extrapolating between predictions conditioned on $𝑦$ and $\hat{y}$, PG effectively **amplifies the exploitation of the indicative prior $y$**,  in the sampling dynamics, thereby improving $p(x_0 | y)$.
>
> **When an additional condition $c$ is present, PG continues to strengthen the utilization of the prior $y$ within $p(x_0 | y, c)$, while remaining orthogonal to how  $c$ is incorporated.**
>
> To further enhance alignment with the additional condition $c$, PG can be naturally combined with CFG. For example, one can train a bridge model to jointly model $p(x_0 | y, c)$ and $p(x_0 | y)$, and then apply CFG to strengthen conditioning on $c$, while PG enhances the contribution of prior information $y$. This complementary combination is also discussed in our paper.
>
> ***
>
> **Response to W2: Hyperparameter tuning**
>
> We thank the reviewer for this important point. As **training-free** guidance methods, PG and its variants introduce design choices **only at inference time**. The **PG guidance scale** controls the strength of prior exploitation and can be tuned similarly to CFG by a simple sweep from a small value to a large value during inference. The **frequency-modulated schedule**, while theoretically motivated by prior exploitation, does not require SNR estimation. In practice, a simple U-shaped schedule is sufficient and works robustly without delicate tuning.
>
> We provide a simple and consistent tuning strategy across tasks and modalities in response to Reviewer YyvJ. Due to character limitations, please kindly visit it.
>
> We will include clearer tuning guidelines in the revision and release **fully open-source implementations** to support reproducibility and future extensions.
>
> ***
>
> **Response to W3: Formatting issues**
>
> We sincerely apologize for the formatting issues in the manuscript and greatly appreciate the reviewer’s careful attention to these details. We will address these issues and thoroughly revise the manuscript to improve its overall readability. We are committed to resolving similar issues throughout the paper, ensuring a higher-quality final version. Thank you again for your valuable feedback.
>
> ***
>
> **Response to Q1: Predictions from previous timesteps**
>
> We thank the reviewer for this insightful suggestion.
>
> Following the reviewer’s idea, we tested using the $x_0$ prediction from the previous time step as the degraded branch, and found that this strategy does work and improves over the DBIM baseline.
>
> Inspired by this, we also explored a bridge-related variant: using the bridge starting point $x_T$ to predict a fixed $x_0$, and then using this fixed prediction as the degraded branch throughout the whole sampling trajectory. This variant also works well.
>
> For both variants, we performed a sufficient search over the guidance scale to ensure that the reported results reflect their best performance.
>
> However, under the same NFE budget, **FMPG still performs the best**. **In particular, for FMPG we already halve the number of sampling steps to keep the total NFE the same, which makes the comparison with the last two guidance variants fair.**. Even under this fair setting, FMPG remains the strongest overall.
>
> | METHOD | NFE=10 | NFE=20 | NFE=40 | NFE=100 |
> | :--- | :--- | :--- | :--- | :--- |
> | DBIM | 4.23 | 3.69 | 3.38 | 3.16 |
> | ECSI | 3.90 | 3.59 | - | - |
> | DBIM+FMPG (Blur) | 3.75 | 3.36 | 3.20 | 3.14 |
> | DBIM+FMPG (Noise) | **3.51** | **3.20** | **3.13** | **3.08** |
> | Prev-step guidance | 3.84 | 3.44 | 3.24 | - |
> | Fixed xT prediction guidance | 3.92 | 3.45 | 3.20 | - |

---

> > ### Author Rebuttal · Reviewer_SLvh · 2026-04-02
> >
> > Thank you to the author for the detailed reply. I would raise my score to 4.

---

> > > ### Author Response · Authors · 2026-04-03
> > >
> > > Thank you for your thoughtful review and insightful questions. We sincerely appreciate your time and recognition of our work. The clarifications from our rebuttal will be incorporated into the revised manuscript.

---

### Official Review · Reviewer_YyvJ · 2026-03-07

**Soundness:** 3
**Presentation:** 2
**Significance:** 3
**Originality:** 3
**Overall Recommendation:** 4
**Confidence:** 3

**Summary:**

This paper presents Prior Guidance (PG), a training-free method for diffusion bridge models that boosts prior exploitation by contrasting denoising results from a learned clean prior and a synthetic weak prior, and extends it to Frequency-Modulated PG (FMPG) which tailors guidance scales to low/high-frequency bands to match bridge models’ U-shaped SNR trajectory, plus a cascaded CFG-FMPG framework for weak-prior tasks like inpainting that uses CFG for early semantic reconstruction and FMPG for late prior exploitation.

**Compliance With Llm Reviewing Policy:**

Affirmed.

**Final Justification:**

The rebuttal has addressed my concerns. I keep my positive score.

**Key Questions For Authors:**

1. How does FMPG perform on high-resolution imagery (512×512/1024×1024)?
2. What is the robustness of PG/FMPG to non-Gaussian weak priors instead of the default Gaussian noise degradation?
3. Is there a way to eliminate task-specific hyperparameter tuning (e.g., guidance scales, corruption σ) for PG/FMPG?

**Limitations:**

yes

**Strengths And Weaknesses:**

Strength:
1. This work delivers the first training-free bridge model-specific guidance framework (PG/FMPG) with rigorous theoretical grounding in bridge models’ U-shaped SNR and frequency exploitation dynamics.
2. It achieves state-of-the-art generation quality and up to 20× sampling acceleration on image tasks, with modular designs that address weak-prior limitations and require no retraining of pre-trained bridge models.
3. Experiments are comprehensive and fair across multiple benchmarks/metrics.

Weakness:
1. The method relies on task-specific hyperparameter tuning with no universal tuning strategy.
2. The experiments are limited to relatively low image resolutions.
3. Most degradation operators for weak prior construction (e.g., JPEG artifacts, 4× pooling) underperform Gaussian noise/blur across most tasks and NFE regimes.
4. FMPG’s frequency decomposition step adds minor computational overhead.

---

> ### Author Rebuttal · Authors · 2026-03-31
>
> **Response to Reviewer YyvJ**
>
> ***
>
> **Response to W1 and Q3: Task-specific hyperparameter tuning**
>
> We thank the reviewer for this insightful question, which helps clarify the practical usability of PG/FMPG. In practice, PG follows a simple and consistent two-stage tuning strategy across tasks.
>
> - Degradation design, where one selects an appropriate degradation form to construct a weak prior. This does not require careful tuning of the degradation scale, as small, structured perturbations are sufficient.
>
> - Guidance scale, which, similar to CFG, PG exhibits an optimal guidance scale. In practice, this can be efficiently found by sweeping the scale from small to large during inference, without task-specific redesign.
>
> While tuning cannot be fully eliminated, it can be reduced to a lightweight, robust, and transferable procedure, as validated across diverse tasks and modalities.
>
> In addition, during the rebuttal, we further evaluated PG on multiple settings (e.g., image-to-image translation, high-resolution images, audio, and video). We will **include clearer tuning guidelines in the revision** and release **fully open-source implementations** for all tasks to support reproducibility and future extensions.
>
> ***
>
> **Response to W2 and Q1: Experiments on high-resolution imagery**
>
> We thank the reviewer for this suggestion. We further validate PG on high-resolution image editing tasks using ViBT (Vision Bridge Transformer at Scale) under its default 16-step bridge sampling setting. PG remains effective across diverse degradation operators, showing it is not limited to a specific corruption type. All experiments use small degradation strengths without task-specific tuning.
>
> **Especially, we keep the original CFG guidance scale unchanged and use PG in the later sampling stage without increasing NFE**. Even under these unchanged settings, PG consistently improves performance, demonstrating its robustness and practical simplicity.
>
> Table 1: Quantitative comparison between PG and the CFG baseline on three high-resolution image editing tasks. We report no-reference image quality metrics (NIQE, TOPIQ, MUSIQ, MANIQA, CLIPIQA), CLIPScore, and VLM-based evaluation scores (IA, EQ, DP, and their average VLM). In this table, the degradation operator is implemented as adding a small amount of noise. Bold indicates the better result for each metric.
> | Task | Method | NIQE ↓ | TOPIQ ↑ | MUSIQ ↑ | MANIQA ↑ | CLIPIQA ↑ | CLIPSc ↑ | VLM-IA ↑ | VLM-EQ ↑ | VLM-DP ↑ | VLM ↑ |
> | :--- | :--- | :--- | :--- | :--- | :--- | :--- | :--- | :--- | :--- | :--- | :--- |
> | Style | CFG | 6.59 | 0.345 | 48.85 | **0.325** | 0.464 | 0.159 | 1.96 | 1.76 | 1.66 | 1.80 |
> | Style | PG | **5.14** | **0.360** | **49.40** | 0.310 | **0.519** | **0.169** | **2.45** | **2.10** | **1.94** | **2.16** |
> | Remove | CFG | 7.31 | 0.333 | 46.48 | **0.336** | 0.465 | 0.163 | 4.18 | 3.58 | 3.24 | 3.67 |
> | Remove | PG | **7.08** | **0.334** | **46.52** | 0.335 | **0.469** | 0.163 | **4.28** | **3.65** | **3.26** | **3.73** |
> | Replace | CFG | **6.92** | 0.313 | 45.03 | 0.328 | 0.509 | 0.192 | 4.42 | 3.60 | 2.90 | 3.64 |
> | Replace | PG | 7.16 | **0.328** | **46.10** | **0.359** | **0.540** | **0.193** | **4.53** | **3.76** | **2.95** | **3.75** |
>
> ******
>
> **Response to W3 and Q2: Robustness to Non-Gaussian weak priors**
>
> We thank the reviewer for the insightful observation. While degradations such as JPEG compression and 4× pooling perform worse than Gaussian noise and blur, **all evaluated degradation types (noise, Gaussian/average blur, JPEG, and 4× pooling) consistently outperform the unguided baseline, demonstrating the robustness of PG across diverse degradations**.
> Empirically, In DBIM setting ,their performance roughly follows: noise > Gaussian/average blur > JPEG/4× pooling.
>
> ***
>
> **Response to W4: Computational overhead resulting from frequency decomposition**
>
> We thank the reviewer for raising this concern. We profile the inference-time overhead of frequency-modulated guidance by measuring the average runtime of the pre-trained denoiser with two guidance strategies: FFT Guidance (used in FMPG) and Scalar Guidance (standard PG).
>
> As shown in Table 2, FFT guidance adds only a small overhead (0.0253s vs. 0.0162s on average when NFE=10), accounting for less than 0.3% of the denoiser runtime. **This indicates that the frequency-modulation step in FMPG introduces negligible inference overhead**.
>
> | Experiment | Denoiser (s) (NFE=20) | FFT Guidance (s)  (NFE=20)| Scalar Guidance (s) | FFT / Denoiser | Scalar / Denoiser |
> | :--- | :--- | :--- | :--- | :--- | :--- |
> | 1 | 9.6865 | 0.0186 | 0.0115 | 0.19% | 0.12% |
> | 2 | 9.6876 | 0.0257 | 0.0201 | 0.27% | 0.21% |
> | 3 | 10.3744 | 0.0395 | 0.0174 | 0.38% | 0.17% |
> | 4 | 9.7197 | 0.0174 | 0.0156 | 0.18% | 0.16% |
> | **Avg** | **9.8671** | **0.0253** | **0.0162** | **0.26%** | **0.16%** |

---

> > ### Author Rebuttal · Reviewer_YyvJ · 2026-04-03
> >
> > The rebuttal has addressed my concerns.

---

> > > ### Author Response · Authors · 2026-04-03
> > >
> > > Thank you very much for your thoughtful review. We are delighted that our rebuttal has addressed your concerns and sincerely appreciate your support. If you feel it is appropriate, we would be grateful if you could consider adjusting your score accordingly. Thank you for your recognition of our work again!

---

### Official Review · Reviewer_4RV4 · 2026-03-11

**Soundness:** 3
**Presentation:** 3
**Significance:** 2
**Originality:** 2
**Overall Recommendation:** 5
**Confidence:** 2

**Summary:**

This paper proposes Prior Guidance (PG), a training-free guidance method specifically designed for diffusion bridge models. The key insight is that bridge models differ from standard diffusion models by starting from informative clean priors rather than pure Gaussian noise, and this prior exploitation capability can be further enhanced through guidance.

**Compliance With Llm Reviewing Policy:**

Affirmed.

**Final Justification:**

The rebuttal has addressed my concerns. I raised my score.

**Key Questions For Authors:**

Q1. The efficiency comparison is currently not fair. Your method uses two network forward passes per sampling step, while the baselines use one. Please report results under matched compute, e.g., compare your method at 20 steps against DBIM at 40 actual network evaluations. Otherwise, it is hard to tell whether the gains come from the method itself or simply from extra computation.

Q2. The analogy to CFG and autoguidance feels weak. In CFG, both branches are seen during training, so the extrapolation is meaningful. Here, the degraded prior is never observed during training, which makes this branch effectively out-of-distribution. Why should extrapolating from such predictions improve sampling rather than amplify artifacts? Some analysis of what the guidance term is actually capturing would make this much more convincing.

Q3. The experimental scope is still narrow. All results are on paired image-to-image tasks, even though bridge models are used in broader settings like video and audio generation. Do PG and FMPG generalize beyond these image tasks? If not, the paper should be clearer about this limitation and discuss what would be required to extend the method.

**Limitations:**

yes

**Strengths And Weaknesses:**

S1. Instead of improving condition alignment like CFG or score estimation like autoguidance, it directly targets prior exploitation, which is arguably the key advantage of bridge models.

S2. The discussion of U-shaped SNR dynamics and frequency-dependent prior accessibility is intuitive and reasonably well supported by the empirical analysis in Figure 3.

S3. Results are reported on multiple datasets and metrics, and the ablations are fairly extensive. The method appears consistently beneficial across different sampling budgets.

W1. The paper relies too much on intuition for several core design choices. In particular, it is still unclear why extrapolating between clean-prior and corrupted-prior predictions should systematically improve prior exploitation rather than just amplify errors. The frequency schedules also feel largely heuristic, despite the physical motivation.

W2. The efficiency claims are overstated. The method requires two denoising passes per sampling step, so the actual compute is roughly doubled. Under equal compute budgets, the claimed acceleration over prior bridge samplers becomes much less clear.

W3. The empirical scope is somewhat narrow. Much of the evaluation is still on relatively limited image-to-image settings, and it is not yet clear how well the method transfers to more challenging or modern bridge-model applications. Generalization beyond the two tested backbones is also left open.

---

> ### Author Rebuttal · Authors · 2026-03-31
>
> **Response to Reviewer 4RV4**
>
> We sincerely thank reviewer 4RV4 for your time and effort in carefully reviewing our manuscript and providing valuable feedback. Below, we provide responses to your insightful comments and concerns.
>
> ***
>
> **Response to W1 and Q2**
>
> PG can be understood as guidance via **denoising quality differences**, similar to CFG and AG, where extrapolation amplifies higher-quality predictions. Our key difference is that **PG constructs this gap along the prior exploitation axis**. The clean prior (seen in training) yields stronger denoising, while **a deliberately degraded prior provides a controlled lower-quality estimate**. The guidance term therefore captures and amplifies the gain from reliable priors, improving prior utilization.
>
> **The degraded prior is not harmful to the result because it serves only as a structured low-quality reference, not a standalone predictor**. The degradation is mild and preserves partial information, ensuring a meaningful contrast signal. Empirically, small perturbations already suffice without careful tuning.
>
> For the frequency schedule, it is clearly motivated by the bridge dynamics. In practice, PG does not require SNR estimation or access to pre-trained bridge models. A simple U-shaped schedule is sufficient and robust, with minimal tuning.
>
> We will revise the paper to clarify these points.
>
> ***
>
> **Response to W2 and Q1: Inference efficiency comparison**
>
> **We ensure strictly fair comparisons under equal compute budgets by aligning the number of function evaluations (NFE) across all methods**. For PG, each step involves two denoising evaluations, so we halve the number of sampling steps to match total NFE with unguided bridge samplers (e.g., DDBM, DBIM). Therefore, the improvements are not due to increased computation, but reflect genuine quality gains.
>
> ***
>
> **Response to W3 and Q3: Empirical scope**
>
> To expand the empirical scope, we further evaluate PG on recent scalable bridge models in the latent space for both audio and video, going beyond image-to-image settings.
>
> - Audio. We adopt AudioLBM (NeurIPS 2025), a state-of-the-art bridge-based audio super-resolution model. Under a training-free setting with strictly aligned NFE (e.g., 50 steps for the AudioLBM baseline vs. 25 steps with PG), PG consistently improves performance on the VCTK dataset. Following the original paper’s protocol, we compare against multiple baselines, AudioLBM, and its VCTK-specific model. The results show that PG effectively enhances a strong bridge model in audio latent space.
>
> Table 1: Audio super-resolution results. PG training-freely improves AudioLBM. The degraded prior is built by interpolating the 8 kHz input $x_1$ with the current state $x_t$, lowering the high resolution content of $x_t$. AudioLBM+PG outperforms both AudioLBM and AudioLBM (VCTK) on key metrics.
>
> | Metric | FlowH | AuSR | AudioLBM | AudioLBM+PG | AudioLBM (VCTK) |
> | :--- | :--- | :--- | :--- | :--- | :--- |
> | LSD↓ | 0.816 | 0.940 | 0.753 | **0.672** | 0.742 |
> | LSD(L)↓ | 0.194 | 0.486 | 0.773 | 0.775 | 0.708 |
> | LSD(H)↓ | 0.889 | 0.994 | 0.724 | **0.639** | 0.712 |
> | SSIM↑ | 0.784 | 0.809 | 0.893 | **0.909** | 0.906 |
> | SigMOS_sig↑ | - | - | 3.673 | **3.707** | - |
> | SigMOS_ovrl↑ | - | - | 3.350 | **3.374** | - |
>
> - Video. We further evaluate on FrameBridge (ICML 2025) for image-to-video (I2V) generation. Again, under training-free and NFE-aligned settings, PG brings consistent improvements, demonstrating that it transfers to more complex spatiotemporal generation tasks.
>
> Table 2: VBench I2V results. We compare prior benchmark results, our reproduced FrameBridge baseline, and the proposed PG method. FrameBridge VideoCrafter denotes the original reported result, and Baseline is our reproduction under the same evaluation pipeline. Minor differences may come from sampling and evaluation stochasticity. Bold indicates the best result, “–” means not reported, and higher is better for all metrics.
>
> | Model / Setting | Total Score | CM | I2V-SC | I2V-BC | SC | BC | MS | DD | AQ | IQ | I2V | I2V_Quality |
> |---|---:|---:|---:|---:|---:|---:|---:|---:|---:|---:|---:|---:|
> | DynamiCrafter-256 | 84.35 | 22.18 | 95.40 | 96.22 | 94.60 | 98.30 | 97.82 | 38.69 | 59.40 | 62.29 | -- | -- |
> | FrameBridge-VideoCrafter | 85.37 | 30.72 | 96.24 | **97.25** | 94.63 | 98.92 | **98.51** | 35.77 | 59.38 | 63.28 | -- | -- |
> | FrameBridge (Baseline) | 85.23 | 30.73 | 95.60 | 96.39 | 93.69 | 98.49 | 95.72 | 33.74 | 58.47 | 63.40 | 92.89 | 77.57 |
> | FrameBridge+PG (blur) | 85.71 (+0.48) | 27.44 | 96.34 | 96.63 | 94.78 | 99.11 | 97.11 | 31.46 | 59.45 | 64.02 | 93.20 | 78.22 |
> | FrameBridge+PG (pooling) | **85.74** (+0.51) | 26.97 | **96.41** | 96.69 | 94.96 | **99.14** | 97.12 | 30.16 | **59.67** | 64.35 | **93.23** | **78.24** |
>
> These results suggest that PG is not limited to image-to-image tasks, but naturally generalizes to other modalities where bridge models are designed, including latent-space, scalable settings.

---

> > ### Author Rebuttal · Reviewer_4RV4 · 2026-04-03
> >
> > Appreciate the response.

---

> > > ### Author Response · Authors · 2026-04-03
> > >
> > > Thank you very much for your comprehensive review and constructive feedback. We truly appreciate that our responses have addressed your concerns.
> > >
> > > In the rebuttal, following your comments and suggestions, we have further clarified the design of our method, provided explanations of its mechanism, illustrated the comparison of inference efficiency, and extended our evaluation to broader settings, including both the video and audio generation scenarios you mentioned. We hope these additions make our contributions more convincing and complete.
> > >
> > > We sincerely appreciate your time and consideration, and if you find it appropriate, we would be very grateful if you could consider adjusting your score accordingly.

---

### Decision · Program_Chairs · 2026-04-30

**Decision:**

Accept (regular)

**Comment:**

I think this is a very interesting manuscript which provides good insights into an important problem. The reviewers are fairly positive about the work and they are also satisfied with the rebuttal provided by the authors. Only Reviewer 3qer is still not completely satisfied but I agree with the authors that the concerns raised by this reviewer have been properly addressed in the rebuttal. Yes, the reviewer is right that the optimal choice of degradation operators requires experience but this is always the case. I like the fact that PG is a simple and training-free approach relying entirely on inference-time experience, avoiding complex conditioning or training design. The approach is shown to be robust across a wide range of tasks like image generation, in-painting, translation, and high-resolution editing, where simple noise addition serves as an effective and often universal degradation operator. The method is rather simple, employing a consistent two-stage tuning strategy: selecting an appropriate degradation operator with minimal effort, followed by a shared guidance-scale search. Overall, I think PG offers a practical and efficient framework that generalizes well across tasks and modalities and this makes is useful to the community at large.